# Accuracy is *Not* All You Need

**Abhinav Dutta**
Microsoft Research
Bangalore, India
t-abdutta@microsoft.com

**Sanjeev Krishnan**
Microsoft Research
Bangalore, India
sakrishnan@microsoft.com

**Nipun Kwatra**
Microsoft Research
Bangalore, India
nipun.kwatra@microsoft.com

**Ramachandran Ramjee**
Microsoft Research
Bangalore, India
ramjee@microsoft.com

## Abstract

When Large Language Models (LLMs) are compressed using techniques such as quantization, the predominant way to demonstrate the validity of such techniques is by measuring the model's accuracy on various benchmarks. If the accuracies of the baseline model and the compressed model are close, it is assumed that there was negligible degradation in quality. However, even when the accuracies of the baseline and compressed model are similar, we observe the phenomenon of *flips*, wherein answers change from correct to incorrect and vice versa in proportion. We conduct a detailed study of metrics across multiple compression techniques, models and datasets, demonstrating that the behavior of compressed models as visible to end-users is often significantly different from the baseline model, even when accuracy is similar. We further evaluate compressed models both qualitatively and quantitatively using MT-Bench and show that compressed models exhibiting high *flips* are worse than baseline models in this free-form generative task. Thus, we argue that accuracy and perplexity are necessary but not sufficient for evaluating compressed models, since these metrics hide large underlying changes that have not been observed by previous work. Hence, compression techniques should also be evaluated using *distance* metrics. We propose two such distance metrics, *KL-Divergence* and *% flips*, and show that they are well correlated.

## 1 Introduction

The high cost and latency of Large Language Models (LLMs) has motivated the design of multiple model compression techniques for optimizing LLM efficiency such as quantization (Dettmers et al., 2022), Key-Value (KV) cache compression (Ge et al., 2023), pruning (Sun et al., 2023) and sparsification (Ashkboos et al., 2024). However, today, there is no standardized way of evaluating the effectiveness of these techniques.

The predominant way of establishing the validity of the LLM compression methods today is to report accuracy on selected benchmark tasks such as MMLU (Hendrycks et al., 2021a), Hellaswag (Zellers et al., 2019), ARC (Clark et al., 2018), LAMBADA (Paperno et al., 2016), etc. It is assumed that if the compressed model preserves accuracy on such benchmarks, it can be used as an equivalent replacement for the baseline model.

In this paper, we conduct a detailed evaluation of various compression techniques. We find that while the difference in the aggregate accuracy metric across various benchmarks between the baseline and compressed LLM is negligible in most cases ( $\leq 2\%$ ), the actual percentage change in the answers

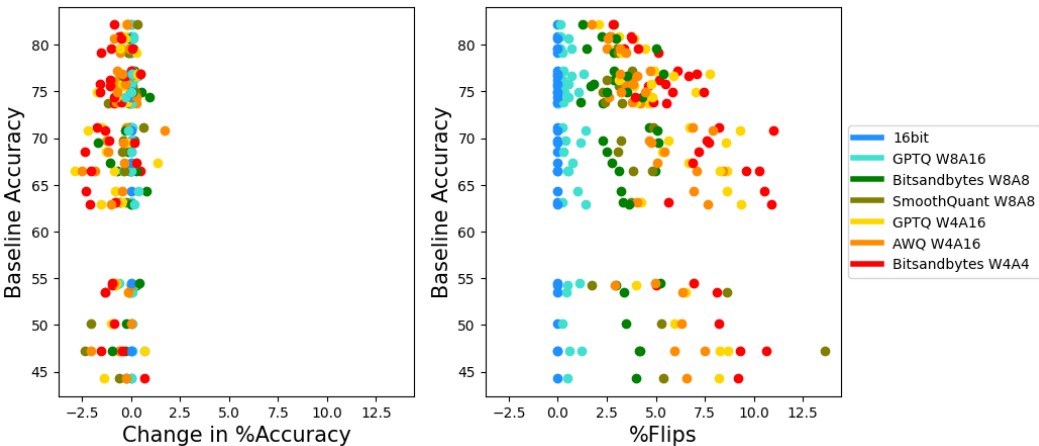

Figure 1: All six quantization schemes show *negligible difference in accuracy* compared to baseline 16-bit model (Llama2-chat 7B, 13B, 70B and Yi-chat 6B, 34B) in seven different tasks. However, all schemes, except GPTQ W8A16 (8-bit weight, 16-bit activation), *exhibit large number of flips*, indicating severe divergence in model behavior.

can be significant ($\geq 5\%$). In other words, even when the overall accuracy is unchanged, a large number of correct answers change to incorrect and vice versa in proportion (we call these *flips*), between the baseline and compressed model. To the best of our knowledge, we believe that *we are the first to identify this phenomenon of flips caused due to model compression. Further, we argue that flips serves as an intuitive metric that captures how significantly different the compressed model is from the baseline model, even when both models exhibit similar accuracy on various benchmarks.*

Figure 1 shows the change in accuracy and flips % vs baseline 16-bit model, respectively, for *six* quantization schemes on *seven* benchmark tasks (MMLU (Hendrycks et al., 2021a), Hellaswag (Zellers et al., 2019), LAMBADA (Paperno et al., 2016), ARC Easy and Challenge (Clark et al., 2018) PIQA (Bisk et al., 2019), and Winogrande (Sakaguchi et al., 2019)). We see that all quantization schemes have negligible difference in accuracy (**0 − 2%**) compared to the 16-bit version. However, except for GPTQ W8A16 (8-bit weight, 16-bit activation Frantar et al. (2023)) that preserves accuracy with negligible flips, all other quantization schemes exhibit large number of flips (**up to 13.6%**), indicating significant divergence from the baseline model.

Figure 3 shows similar behavior of MMLU task accuracy being preserved while flips increase, for two other compression techniques, namely, layer dropping (Gromov et al., 2024) and WANDA weight pruning (Sun et al., 2023). For example, while Gromov et al. (2024) showed that dropping the last few layers of a model did not affect its accuracy on standard benchmarks, we find a steady, almost linear increase in the number of flips with the number of layers being dropped.

The phenomenon of flips is puzzling at first glance. While it is easy to see that some correct answers may become incorrect due to errors induced by compression, it is difficult to explain how an approximately equal number of incorrect answers become correct such that overall accuracy is preserved! For example, MMLU questions have 4 options, one of which is correct. Thus, any output change could move a correct answer to an incorrect one but there is only 1 in 3 chance for an incorrect answer to land on the correct option. We present a detailed analysis of flips in Section 5. Furthermore, we observe that simply adding Gaussian noise to model weights can reproduce this flips phenomenon (see Table 1). This suggests flips arise from the inherent approximations introduced by compression rather than any new information or learning during the compression process.

Finally, one might question whether flips matter if accuracy is preserved. Indeed, if the downstream task where the LLM is used closely matches the benchmark task, accuracy alone might suffice. However, LLMs are typically used in a variety of downstream tasks that require generating free-form text, where accuracy evaluated on some standard question-answering tasks could be a poor proxy. Thus, we evaluate the compressed models using MT-Bench (Zheng et al., 2023), a multi-turn dialogue task. We show through qualitative evaluation as well as using GPT4 as an automated judge that

Table 1: Adding Gaussian noise to weights results in *approximately equal correct $\rightarrow$ incorrect* and *incorrect $\rightarrow$ correct* transitions, with the overall model accuracy mostly unchanged.

| Task | Llama3-8b Setting | %Accuracy | % Flips |
|---|---|---|---|
| GSM8k | No noise | 49.39 | - |
| | Noise std = $10^{-4}$ | 48.97 | 8.23 |
| ARC-challenge | No noise | 53.24 | - |
| | Noise std = $5 \cdot 10^{-4}$ | 53.49 | 6.05 |

compressed models with high number of flips *are significantly worse than baseline models* in this task (see Section 6).

Since the goal of compression schemes is to create models that mimic the baseline models as closely as possible, we argue that compressed models are better judged by *distance metrics* with respect to baseline, in addition to *capability metrics* such as accuracy alone, as is the practice today. We demonstrate that well-known distance metrics like *KL-Divergence* on a given dataset can better identify the differences created due to various compression techniques and this metric correlates well with *flips*. Further, we show that the scores on MT-Bench (which evaluates free-form generation capabilities of these models) is highly correlated with *flips*. Thus, we propose that *flips*, an intuitive and inexpensive to compute metric, as a potential proxy distance metric for evaluating LLM compression techniques.

In this paper, we make the following key contributions:

- Using detailed qualitative and quantitative evaluation of various compression techniques, we show that accuracy is not sufficient as an evaluation metric for LLM compression techniques.

- We demonstrate the existence of *flips* as a general phenomenon and explain why they occur.

- We evaluate compression techniques using the *KL-Divergence* distance metric and show that it correlates well with *flips*.

- We propose that, where appropriate, *flips* be used as an intuitive distance metric for evaluating the quality of compression techniques.

## 2   LLM Evaluation Metrics

We compare baseline and compressed LLMs on the following metrics:

- **Accuracy** - *capability* metric: % correct answers, for question-answering tasks. This determines the competency of the model for a particular task. Multiple-choice question-answering (MCQ) tasks such as MMLU expect the model to output a single token for the correct answer (A/B/C/D), and compare this token with the target answer. For other tasks (like PIQA, Hellaswag, ARC), where the model assigns a probability to an option (consisting of multiple tokens), we report the standard *normalized* accuracy (Eleuther, 2021).

- **Perplexity**(Jelinek et al., 2005) - *capability* metric: This measures the overall language modelling capability of an LLM. It is defined as $e^{(Average\ Negative\ Loglikelihood)}$ calculated over a dataset.

- **Flips** - *distance* metric: measures the % of questions whose answers changed from correct $\rightarrow$ incorrect or incorrect $\rightarrow$ correct, between baseline and quantized model for all tasks that have correct/incorrect answers. Note that, we do not include incorrect $\rightarrow$ incorrect transition in Flips for two reasons: 1) For non-MCQ tasks such as GSM8k (Cobbe et al., 2021b), TriviaQA (Joshi et al., 2017), etc. exact per-token output matches between different models are rare, resulting in many mismatches. Thus, including this transition may artificially inflate the metric for these tasks. 2) For MCQ tasks, users may care less about these incorrect $\rightarrow$ incorrect transitions. Nevertheless, *if we include incorrect $\rightarrow$ incorrect transitions for MCQ tasks, we find that, the flips numbers reported in this paper would further increase by another 20-40% (e.g., increase of 19% in Hellaswag, 41% in ARC and 43% in MMLU! See Table 11)*

- **KL-divergence**(Kullback and Leibler, 1951) - *distance* metric: consider a dataset having samples with multiple-choice answer options , where the j-th token of the i-th answer option has a probability distribution $P_b(i,j)$ across all tokens in the vocabulary of the baseline model, and $P_q(i,j)$ for the quantized model. Then the KL-divergence between the models for the entire dataset is the mean of KL-divergences across all tokens of all answer options and all samples in the dataset.

$$KL\ div = \frac{1}{N} \sum_{dataset} \frac{1}{|options|} \sum_{i \in options} \frac{1}{|tokens|} \sum_{j \in tokens} D_{KL}(P_b(i,j)||P_q(i,j)) \quad (1)$$

where N is the number of samples in the dataset and $D_{KL}(P||Q)$ is the standard KL-Divergence between two probability distributions.

The flips metric is propitious because it is a proxy distance metric that is easily interpretable by end-users– for question-answering tasks, the end user typically cares about the correct/incorrect answers and not the underlying probability distribution of tokens. Further, the flips metric is as easy to calculate as accuracy for any dataset.

It is important to distinguish between *capability* metrics (accuracy and perplexity in this study) and *distance* metrics (*KL-Divergence* and *flips* in this study). This distinction is necessary because the goal of a compression scheme is to create a more efficient model that closely mimics the baseline model rather than to create a more capable model. In other words, a quantized model is intended to serve as a drop-in replacement for the baseline model with minimal impact on end-users. Therefore, we argue that *distance* metrics are more suitable for judging the effectiveness of quantization or other compression schemes.

## 3   Experiments

We have measured the above metrics on multiple LLMs using multiple quantization techniques and bit lengths, on several tasks, as listed below:

- Models: We primarily used the Llama2 (7B, 13B, 70B) chat (Touvron et al., 2023), Yi (6B, 34B) chat (01.AI et al., 2024), Llama3 (8B, 70B) (Dubey et al., 2024) and Qwen2 (1.5B, 7B, 72B) (Yang et al., 2024) families of models. The chat versions were used because they can be evaluated on MT-Bench (Zheng et al., 2023). However, we have observed a similar phenomenon in their pretrained non-chat versions as well (see Table 12).

- Quantization: We have evaluated LLM.int8() (Dettmers et al., 2022) as implemented in Bitsandbytes (Dettmers, 2024), with its 8-bit and 4-bit versions (referred to as BnB W8A8 and BnB W4A4 respectively) with default parameters supported with HuggingFace Transformers (Wolf et al., 2020). We used GPTQ (Frantar et al., 2023), AWQ (Lin et al., 2024) with group-size 128 with other parameters being default. We used Smoothquant (Xiao et al., 2024) (referred to as SQ W8A8) with per-token, per-channel quantization using $\alpha = 0.5$. We use TensorRT (NVIDIA, 2024) for SmoothQuant, all other schemes were evaluated using HuggingFace Transformers.

- Tasks:
  1. For the Llama2 and Yi families, we evaluate the compressed models on ten different tasks. They include MMLU (Hendrycks et al., 2021a) Table 4, ARC (Clark et al., 2018)(easy Table 7 and challenge Table 8), PIQA (Bisk et al., 2019) Table 5, Winogrande (Sakaguchi et al., 2019) Table 10, Hellaswag (Zellers et al., 2019) Table 6, and Lambada (Zellers et al., 2019) Table 9. We also use GSM8k (Cobbe et al., 2021a) Figure 13, TriviaQA Joshi et al. (2017) Figure 14 and MT-Bench (Zheng et al., 2023) to evaluate models on generative tasks. MT-Bench is a dataset with 80 two-turn questions which can test generative capabilities of a model. In this study, we have used GPT-4 (OpenAI et al., 2024) (v0314) as judge, to generate the scores reported in Table 2.
  2. For the Qwen2 and Llama3 families, we evaluate on MMLU Table 17, GSM8k Table 21, ARC (easy Table 18, challenge Table 19), MATH (Hendrycks et al., 2021b) Table 20, BFCL (Yan et al., 2024) Figure 23, and Scrolls-Quality (Shaham et al., 2022) Table 22

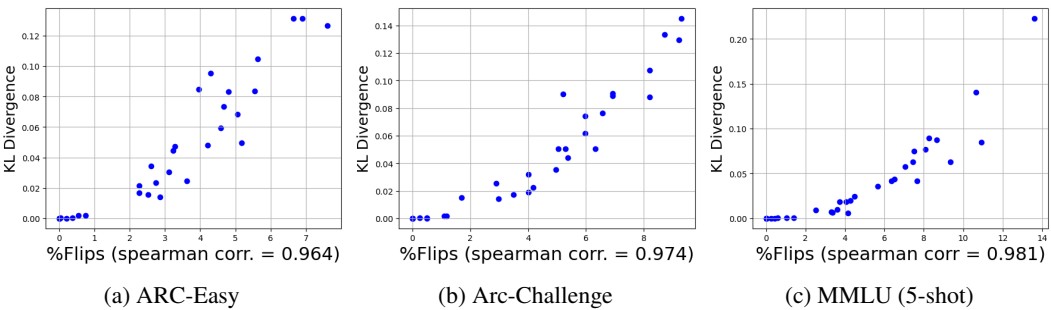

Figure 2: Flips and KL Divergence are well correlated. Each point corresponds to a model, quantization combination in Table 4

- Harness- We used Eleuther AI's eval-harness (Gao et al., 2023) for all the experiments, unless specified otherwise. Note that the standard benchmarks (ARC, MMLU, PIQA, Hellaswag, LAMBADA, GSM8k, TriviaQA, MATH, etc.) results on all models use greedy decoding, making these results fully deterministic.

# 4   Results

In this section, we present extensive evidence for flips across various quantization and pruning schemes, evaluated over a large number of models and all tasks. Results for MT-Bench are presented in Section 6.

## 4.1   Quantization schemes

Summary of our results is highlighted in Figure 1 while the performance on each of the individual seven tasks (MMLU, PIQA, Hellaswag, ARC Easy, ARC Challenge, LAMBADA and Winogrande) are in Tables 4 to 10, respectively, in the Appendix.

The main observations from our experiments with quantized models can be summarized as follows:

1. **Accuracy:** Accuracy is preserved within 1% for the majority of the quantization methods, tasks and models (see Tables 4- 9). This indicates that accuracy is not sufficient to distinguish between precise and permissive quantization schemes.

2. **Flips:** The large %flips ($\geq 5\%$) is a general trend, which holds over different models, almost all quantization schemes, and tasks (see Tables 4- 10). Specifically, all quantization schemes except GPTQ W8A16 ($\leq 1\%$) have significant %flips . Lower bit quantization schemes have greater %flips in general, indicating greater difference in behavior from the baseline (for example, on MMLU, BnB W4A4 has, on average, 2.4× more flips than BnB W8A8). We focus on Flips in this study, but *AllFlips (Flips + incorrect→incorrect transitions )* results can be found in Figure 10, and Table 11 in Appendix.

3. **KL-Divergence vs Flips:** From Figure 2, we observe that the two distance metrics *KL-Divergence* and *%flips* are well correlated. For example, their Spearman correlation on the MMLU benchmark is 0.981.

4. **Impact of task type:**
   - *MCQ* Tasks – Generally easier tasks (identified by higher average accuracy) have smaller %flips. For example, MMLU which is a relatively hard task has 8-16% flips for Bitsandbytes W4A4 whereas for the same technique, PIQA, an easier task, has 3-6% flips. The reason for this behavior is explained in Section 5.
   - *Generative* Tasks – Surprisingly, such tasks have much *more flips* than MCQ ones. For example, GSM8K (Table 13, Table 21), a hard task that requires reasoning over multiple steps, exhibits a significant amount of *flips* (10–25% for BnB W8A8 and W4A4). Similarly, in MATH Table 20, we observe 5-15% *flips* for BnB W4A4. However, *flips* are quite small (2–4%) in easier tasks like TriviaQA(Table 14) that tests trivia question answering capabilities.

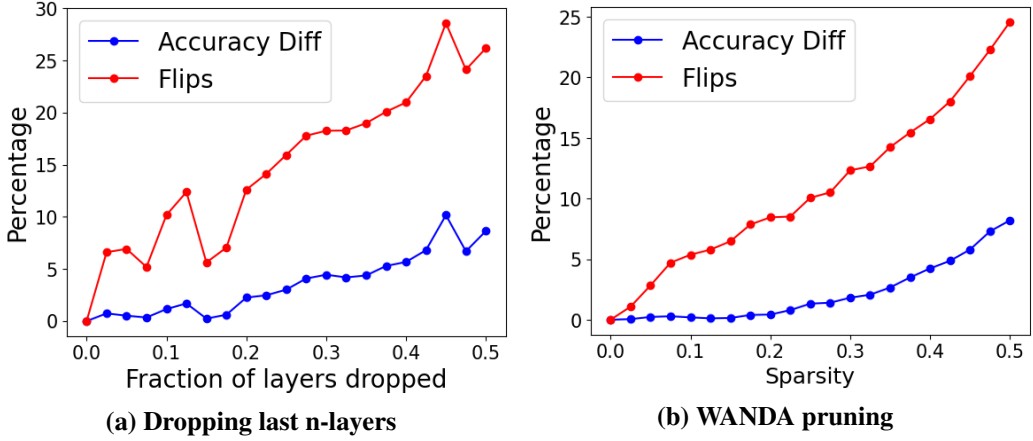

**(a) Dropping last n-layers**          **(b) WANDA pruning**

Figure 3: MMLU 5-shot accuracy difference and flips for two compression techniques (Llama2-13b model). Even at early stages of pruning with no accuracy difference, flips indicate model divergence.

5. **Impact of model size:** Larger models typically have fewer flips than smaller ones. For example, on the MMLU benchmark with BnB W4A4, Llama2-70b chat has 5.6% flips, while Llama2-13b chat and Llama2-7b chat have 1.4× and 1.6× more flips, respectively. This may be because larger models are more resistant to perturbations introduced by compression than smaller ones.

## 4.2 Other model compression techniques

We also evaluate the following three compression techniques, though on a smaller set of tasks and models. Our general observations seen above holds.

1. **Dropping last n-layers** (Gromov et al., 2024): This work demonstrated that dropping the last few layers did not affect the accuracy on standard benchmarks. We find in Figure 3(a) that as one keeps dropping layers, even though the accuracy increases only modestly, %flips increases significantly, demonstrating that the resulting models keep deviating further away from the baseline.

2. **Wanda** (Sun et al., 2023): This is a pruning method. We observe in Figure 3(b) that as we increase the pruning ratio, even though accuracy barely changes, %flips increases steadily.

3. **SliceGPT** (Ashkboos et al., 2024): This is a model sparsification method which drops a certain fraction of rows and columns of each dense matrix. We observe in Figure 9 in Appendix that even at very low sparsity ratios %flips is significant indicating that the compressed models are probably very different from baseline.

## 4.3 Perplexity

Though we have focused on accuracy so far, our observation that the difference between two models' output token values cancel out leaving the average metric result unchanged, is applicable to perplexity as well. In particular, since perplexity may be interpreted as the inverse of the geometric mean of token probabilities, lower probabilities for some tokens in the test dataset may be cancelled by higher probabilities of other tokens. This indicates that perplexity alone is also inadequate in evaluating model compression schemes. Therefore, we argue that along with perplexity, KL-Divergence between the distributions generated by the baseline and optimized models should also be reported.

Figure 11 in Appendix plots the log-likelihood difference between the 16-bit and quantized model for each of the tokens in the wiki-2 dataset (Merity et al., 2016) for four different quantization schemes. From the figure, it appears that the log-likelihoods of the quantized model is just the log-likelihood of the baseline model with some symmetric noise added. Now, since perplexity is $e^{-avg(logprobabilities)}$, adding *any* amount of symmetric noise leaves it unchanged. For example, addition of Gaussian noise to the log-probability outputs of the model maintains the perplexity, while

the quality of generation degrades as the standard deviation of the added noise increases (see Table 29). This analysis demonstrates one key weakness with the perplexity metric when used for evaluating compression techniques. While it is not clear if adding Gaussian noise to the log-likelihoods is an accurate representation of the behavior of compression schemes, it appears to be a reasonable proxy. As we shall see in Section 6, as quantization increases, there is steady degradation in the quality of the text generated by the model that are visible only by examining them closely.

# 5   Analyzing Flips

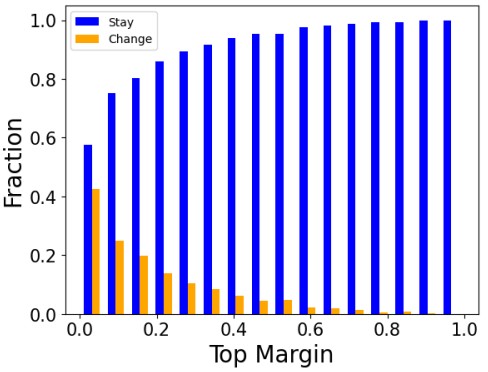
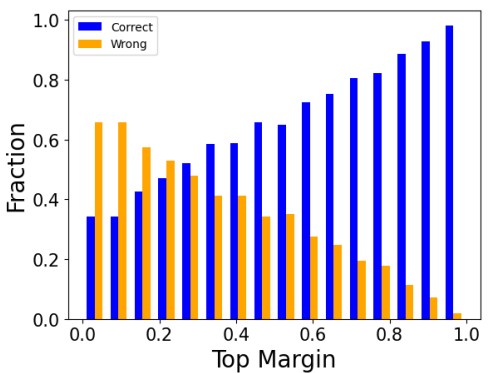

Figure 4: When the Top Margin is low, answer will more likely change (Llama2-70b, BnB W4A4, MMLU 5-shot)

Figure 5: When the Top Margin is low, answer will more likely be incorrect (Llama2-70b, MMLU 5-shot)

One of the interesting observations in this study has been that when we quantize models, the number of questions where the LLM's answers go from incorrect to correct (referred to as $incorrect \rightarrow correct$) is roughly equal to the number that goes the other way. This may seem unintuitive, because one might expect $correct \rightarrow incorrect \gg incorrect \rightarrow correct$, since a) the number of questions with correct answers is usually greater than incorrect answers, so random perturbations should cause more correct answers to flip, and b) given a correct answer, the correct to incorrect transition should be likelier because changing to any of multiple other incorrect options suffices, but given an incorrect answer, the incorrect to correct transition happens only if somehow the perturbation caused by quantization helps it land on the one correct option out of many. But we observe that this is not the case (and indeed, the opposite may also be true in some cases!).

To help explain the above phenomenon, we introduce a metric called *top margin* which is the difference in token probability between the best and the second best answer option. By best (second-best) option, we mean the option that was given the highest (second highest) probability. Higher top margin on a question indicates that the model is more confident about its answer.

**Answers are likely to change when top margin is low.** Quantization introduces some noise in the weights and activations, due to which there is a perturbation in the output answers' probabilities (verified empirically). Thus, we expect that answers are more likely to change when top margin is low, since a small increase or decrease in probabilities can cause the best and second best options to swap (see Figure 4). To further bolster this claim, we show that the changes in probabilities do not depend on top margin, i.e., roughly all questions undergo the same amount noise (except when the top-margin is very high, where we do not see much change in probabilities after compression, but such questions are not likely to flip anyway) as seen in Figure 7. We further find top margins are well correlated before/after compression (i.e low confidence answers are likely to remain so and vice-versa) in Figure 8. In subsection A.4 we find that due to this reason, it is very likely that the same question (with low top margin) would be flipped by multiple quantization schemes.

**Correct (incorrect) answers have higher (lower) top margin and are thus less (more) likely to flip.** Table 27 shows the top margins for questions for which the LLM's answer is correct and when the answer is incorrect. We observe that, top margin when correct is, on average, greater than the top

Table 2: MT-Bench: Average of turn-1 and turn-2 scores, as evaluated by GPT4

| Model | 16bit | BnB W8A8 | GPTQ W8A16 | SQ W8A8 | GPTQ W4A16 | AWQ W4A16 | BnB W4A4 |
|---|---|---|---|---|---|---|---|
| Llama-2 7b chat | 6.375 | 6.375 | 6.384 | 6.377 | 6.018 | 6.015 | 6.317 |
| Llama-2 13b chat | 6.515 | 6.540 | 6.515 | 6.862 | 6.459 | 6.443 | 6.806 |
| Llama-2 70b chat | 7.431 | 7.059 | 7.225 | 7.003 | 6.801 | 6.937 | 7.018 |
| Yi-6b chat | 6.187 | 5.937 | 6.087 | NA | 5.751 | 6.096 | 5.840 |
| Yi-34b chat | 7.387 | 7.220 | 7.337 | NA | 7.156 | 7.053 | 7.185 |

margin when incorrect. This is demonstrated in Table 28 which shows that flips amongst incorrect answers is indeed higher by $2\times$ or more. Similarly, Figure 5 also shows that when top margin is low, the answer is more likely incorrect. Thus, correct answers flip much less often than incorrect answers.

For incorrect answers, we would expect roughly 33% chance of them ending correct (for 4-choice MCQ), though the actual % is typically higher because all the remaining options are not equally likely. Thus, the combination of incorrect answers flipping more along with slightly higher odds than random in landing on the correct answer results in $incorrect \rightarrow correct$ transitions roughly matching $correct \rightarrow incorrect$ transitions.

In subsection A.6 we provide some additional empirical analysis for flips, especially for generative tasks where the above top-margin analysis does not strictly apply.

# 6 MT-Bench evaluation

In this section, we use MT-Bench (Zheng et al., 2023) to evaluate the quantized models' free-form text generation capabilities, using both GPT4 as judge as well as through manual inspection of the model responses.

We first use GPT-4 as a judge and perform automated evaluation. Table 2 shows the MT-Bench average scores for the two turns in the benchmark (individual turn 1 and 2 scores can be found in Tables 15 and 16 in the Appendix). From the results, we can observe that

- Most quantization methods degrade the MT-Bench score for the larger models, by 5% for Llama2-70b chat and 1.5% for Yi-34b chat (Table 2).
- The degradation in MT-Bench score is higher for the harder turn-2 problem than for turn-1, with up to 10% loss for Llama2-70b chat and 5% for Yi-34b chat(Table 16).
- Some quantization methods do slightly better than the baseline in MT-Bench score for smaller models but given their lower overall absolute score, we believe this variation is likely caused by the inaccuracies in GPT4 evaluation process.

For the different compressed models, we compare them on flips in MMLU vs their difference from baseline on MT-Bench scores in Figure 6a. For larger and more capable models, we find that flips in MMLU correlates well with MT-Bench score.

## 6.1 Qualitative evaluation

Next, we perform a detailed qualitative examination of the performance of these models. Specifically, we choose the Llama2-70B-chat model since it has the highest MT-Bench score (Table 2). We compare the 16-bit baseline against 8-bit and 4-bit models, quantized using LLM.int8(). We chose LLM.int8() as it matches the accuracy of the baseline on most tasks and also has the highest GPT4 scores among the W8A8 and 4-bit quantized models for this task (Table 2).

We summarize our findings of the qualitative analysis for a sample of ten questions (out of $\approx$ 30 that had similar issues) from MT-Bench in Table 3. The corresponding generated text of all three models for these questions are provided in Table 34. Overall, we find that the 4-bit and 8-bit models are *significantly worse* than the 16-bit baseline. Specifically, we find that *the 4-bit model often does not follow the provided instruction, makes more mistakes, and rambles a lot more*, with the 8-bit model performing in-between the 16-bit and 4-bit models.

We encourage the reader to look at the full model responses in Table 34 (at least the first one!) to convince themselves that, at least for this task, there is significant degradation due to quantization,

Figure 6: *Flips* is a better predictor of downstream task performance than *Accuracy*

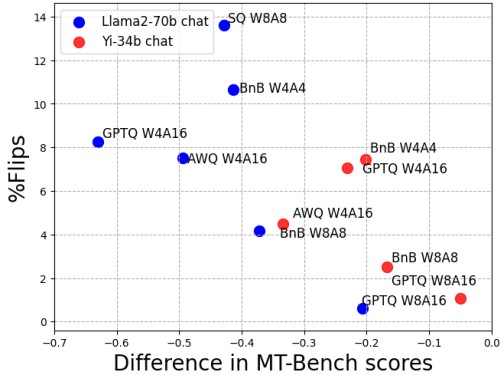 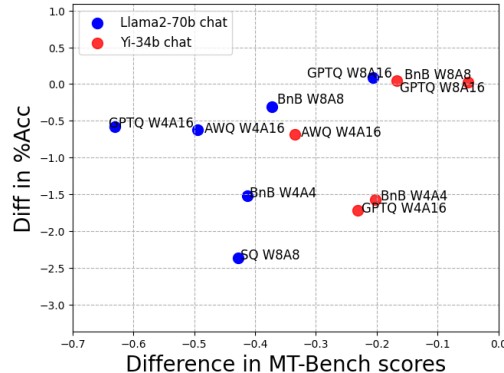

(a) Models with higher flips on MMLU get lower MT-Bench score. Spearman Corr. = -0.75

(b) Models with lower accuracy usually get lower MT-Bench score though the relationship is not as clear. Spearman Corr. = 0.36

Table 3: Qualitative evaluation of Llama2-70B-chat model text generations for MT-Bench prompts. Author's summary of model responses shown below; full model generated responses are in Appendix. These results substantiate a clear degradation in response quality with quantization.

| MT-Bench Prompt | Summary of 16-bit, 8-bit (BnB W8A8), and 4-bit (BnB W4A4) Llama-2-70B-chat model responses |
|---|---|
| 1) Consider a satellite that is in a circular orbit around the Earth. The speed of the satellite decreases. What will happen to the satellite's orbital radius and period of revolution? Please justify your answer using principles of Physics. | 1) Only the 16-bit answer and explanation that radius and revolution period will increase is correct, 8-bit and 4-bit answer that radius will decrease and revolution period will increase/remain constant, respectively, and justify their answers based on (incorrect) Physics! |
| 2) Take your previous response and rephrase it as a limerick. | 2) 16-bit is correct, 8-bit is not a limerick, 4-bit is a limerick but unsound (uses hump and bump for phone). |
| 3) Could you write a captivating short story beginning with the sentence: The old abandoned house at the end of the street held a secret that no one had ever discovered. | 3) 4-bit does not follow the instruction of starting the story with the given sentence. The 16-bit story is more realistic than the 8-bit/4-bit ones. |
| 4) You can see a beautiful red house to your left and a hypnotic greenhouse to your right, an attractive heated pink place in the front. So, where is the White House? | 4) 16-bit is correct. 8-bit says White House is not in your line of sight and towards your back, 4-bit says White House is in the middle! |
| 5) What about when twice the number is divided by 5? | 5) 16-bit and 4-bit are correct, 8-bit is incorrect. |
| 6) Reformulate your earlier reply, output it in JSON format and only include books published after 1980. | 6) 16-bit and 8-bit are correct, 4-bit includes books from 1954 but not 1997! |
| 7) Can you change the ratings from numbers to letters? Capital letters MUST be used when writing the names of phones. | 7) No model follows the Capital letters instruction. 4-bit further messes up, changing a rating of 8.2 to B and a rating of 8.0 to B+! |
| 8) Given a set of complex equations, extract all unique variable names from each equation... | 8) 16-bit is correct, 8-bit and 4-bit think pi is a variable |
| 9) Rewrite your previous response. Start every sentence with an A. | 9) 16-bit follows correctly, 8-bit less fluent, 4-bit is a collection of sentences and makes the mistake of capitalizing the *second* word in every sentence! |
| 10) What is the central dogma of molecular biology? What processes are involved? Who named this? | 10) 16-bit lists four points, 8-bit reproduces the first three of the 16-bit, 4-bit lists the first two points of the 16-bit, indicating steady quality degradation with quantization. |

despite these two compressed models matching baseline accuracy on various tasks (e.g., MMLU accuracy within 1%) and suffering only a 0.4 lower score on a scale of ten in the GPT4 evaluation. We believe that this qualitative analysis adds further evidence to our claim that benchmark accuracy alone, as is standard practice today, is a poor metric to evaluate compressed LLMs, especially, if they are likely to be used for generative tasks in downstream applications.

# 7   Limitations

Predicting performance degradation of LLMs in the wild is a challenging and open problem, and it is possible that *any* metric calculated on standard benchmarks is insufficient. Other limitations are:

- If the downstream task is very similar to the benchmark on which the quantized model is tested, then accuracy may be sufficient, and distance metrics are not needed.

- Flips is only a warning that the behaviour of a model and its compressed version is different – this may or may not materialize as visible degradation in some downstream tasks.

- Our qualitative evaluation in Section 6.1 is subjective and may not be broadly representative.

# 8   Related Work

Given their versatility, LLMs are evaluated on a diverse set of tasks (Chang et al., 2024). Since accuracy is one of the most well-accepted metrics used in task evaluation, compression methods today typically focus on accuracy. However, we are not the first to point out the problem with over-reliance on aggregate metrics like accuracy when judging the quality of a model optimization scheme. Xu et al. (2021) have proposed label loyalty and probability loyalty as a metric to evaluate compressed BERT models . Other works like Joseph et al. (2021), Hooker et al. (2020), and Hooker et al. (2021) have shown compressed ImageNets to be more biased despite preserving accuracy and have proposed Knowledge Distillation based methods to address it. There has also been work (Hong et al., 2024) on evaluating LLM compression schemes on various trustworthiness dimensions. However, metrics for evaluating LLM compression techniques have not been studied widely so far, leading to over reliance on accuracy alone.

There have been many works on LLM evaluation that have shown shortcomings of existing evaluation methods. Lyu et al. (2024) have pointed out the misalignment between free-form generation and probability based evaluation on MMLU. Sclar et al. (2023) have shown LLMs to be very sensitive to prompt formatting. Zheng et al. (2024) have shown models to be biased towards a certain option in MCQ tasks. Alzahrani et al. (2024) have shown minor changes in the benchmarks leading to re-ordering of rankings, and Srivastava et al. (2024) has shown accuracies to be different when considering the *functional* equivalent of math problems. Jaiswal et al. (2024) have curated existing datasets to create their own benchmark that can be used to evaluate compressed models. Li et al. (2024) and Jin et al. (2024) have evaluated various quantization tasks on multiple tasks. Namburi et al. (2023) have studied the impact of compression and pruning on an LLM's *parametric* knowledge. Zhang et al. (2024) propose a number of other metrics in addition to accuracy such as fluency, informativeness, coherence and harmlessness. Chang et al. (2024) presents a detailed survey on evaluation of LLMs that covers what, where, and how to evaluate an LLM and lists several challenges in LLM evaluation.

However, to the best of our knowledge, none of the prior work have pointed out the phenomenon of flips, that occurs when LLMs are compressed, and the observation that higher flips is correlated with larger degradation in model performance despite accuracy matching with the uncompressed model.

# 9   Conclusion

In this work, we have examined metrics to evaluate the quality of compression methods for LLMs such as quantization. We distinguish between aggregate capability metrics such as accuracy, and distance metrics *flips* and *KL Divergence* between the compressed model and the baseline model. We justify why using distance metrics is more appropriate for evaluating model compression methods. We show that accuracy severely underestimates the true distance between models as perceived by the end user. We explain this is due to the presence of flips between correct and wrong answers when a model is quantized, and explain why the flips are nearly balanced, leading to similar accuracy, while the user-perceived output of the quantized model may be significantly different. We argue that distance metrics such as flips and KL-divergence are essential for evaluating all optimization methods which may change the model outputs and whose goal is to minimize end-user visible behaviour changes from a baseline model. We hope that better distance metrics as proposed in this work will enable research in model optimization and compression to progress faster and better meet user expectations on model output quality.

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

# A    Appendix

**Outline**

## A.1    Detailed results of Llama-2 chat and Yi-chat Families on various quantization schemes

Table 4 shows five-shot accuracy on the for various models using the standard 16-bit quantization as baseline and difference in accuracy and percentage of flips for various lower-bit quantization schemes. For example, the accuracy of Bitsandbytes (Dettmers et al., 2022) 8-bit and 4-bit quantized models are only 0.55% and 0.78% away from the baseline Llama2-70b model respectively while the flips are 4.6% and 8.1%, respectively. Tables 5 through 10 show results for other zero-shot tasks such as PIQA, Hellaswag, ARC Easy, ARC Challenge, LAMBADA and Winogrande.

Table 4: MMLU 5-shot accuracy and flips for several models using 16-bit baseline and various quantization schemes. Change in accuracy is negligible (0-2%) in all quantization schemes. However, except for GPTQ W8A16 (8-bit weights, 16-bit activation), all other schemes show large % flips, indicating significant deviation of quantized model from the baseline 16-bit model.

| Model | 16-bit Baseline Accuracy (%) | BnB W8A8 | GPTQ W8A16 | SQ W8A8 | GPTQ W4A16 | AWQ W4A16 | BnB W4A4 |
|---|---|---|---|---|---|---|---|
| | | Change in accuracy/ flips (%), compared to 16-bit baseline | | | | | |
| Llama2-7b chat | 47.21 | -0.30 / 4.15 | 0.08 / 0.60 | -2.36 / 13.62 | -0.58 / 8.26 | -0.62 / 7.50 | -1.51 / 10.65 |
| Llama2-13b chat | 53.54 | -0.10 / 3.35 | 0.07 / 0.43 | -1.34 / 8.65 | -0.14 / 6.49 | -0.15 / 6.36 | -1.31 / 8.09 |
| Llama2-70b chat | 63.17 | -0.55 / 3.32 | -0.01 / 0.26 | -0.24 / 3.73 | -0.45 / 4.26 | -0.76 / 4.05 | -0.78 / 5.65 |
| Yi-6b chat | 62.95 | 0.00 / 3.62 | 0.15 / 1.40 | NA / NA | -1.53 / 9.36 | -1.00 / 7.66 | -2.07 / 10.90 |
| Yi-34b chat | 74.89 | 0.05 / 2.51 | 0.03 / 1.05 | NA / NA | -1.71 / 7.05 | -0.68 / 4.47 | -1.57 / 7.44 |

Table 5: PIQA (0-shot) change in %accuracy / %flips

| Model | 16bit | BnB W8A8 | GPTQ W8A16 | SQ W8A8 | GPTQ W4A16 | AWQ W4A16 | BnB W4A4 |
|---|---|---|---|---|---|---|---|
| Llama2-7b chat | 77.203 | 0.16 / 2.88 | 0.05 / 0.27 | 0.00 / 3.80 | 0.00 / 4.87 | -0.70 / 4.73 | 0.00 / 6.09 |
| Llama2-13b chat | 79.162 | -0.16 / 2.88 | 0.00 / 0.21 | -0.11 / 3.04 | 0.27 / 3.21 | -0.54 / 3.48 | -1.52 / 5.11 |
| Llama2-70b chat | 80.903 | -0.49 / 2.23 | 0.11 / 0.108 | -0.49 / 2.66 | -0.27 / 3.10 | -0.54 / 2.72 | -0.60 / 3.75 |
| Yi-6b chat | 76.659 | -0.38 / 3.53 | 0.38 / 0.707 | NA / NA | 0.21 / 5.87 | 0.00 / 4.03 | -0.27 / 6.69 |
| Yi-34b chat | 79.597 | -0.54 / 5.01 | -0.05 / 0.71 | NA / NA | -0.54 / 4.46 | -0.11 / 3.16 | 0.05 / 4.08 |

Table 6: Hellaswag (0-shot) change in %accuracy / %flips

| Model | 16bit | BnB W8A8 | GPTQ W8A16 | SQ W8A8 | GPTQ W4A16 | AWQ W4A16 | BnB W4A4 |
|---|---|---|---|---|---|---|---|
| Llama2-7b chat | 75.532 | -0.03 / 1.66 | 0.05 / 0.29 | 0.06 / 3.13 | -0.40 / 3.88 | -0.66 / 3.47 | -1.06 / 4.63 |
| Llama2-13b chat | 79.635 | -0.10 / 1.49 | 0.00 / 0.14 | 0.13 / 3.00 | -0.45 / 2.58 | -0.54 / 2.48 | -0.99 / 3.38 |
| Llama2-70b chat | 82.164 | -0.17 / 1.26 | -0.02 / 0.12 | 0.31 / 2.83 | -0.22 / 2.11 | -0.19 / 1.74 | -0.84 / 2.80 |
| Yi-6b chat | 75.771 | -0.14 / 1.81 | 0.0 / 0.56 | NA / NA | -0.45 / 4.79 | -0.31 / 3.81 | -1.56 / 5.51 |
| Yi-34b chat | 80.681 | -0.15 / 2.98 | 0.10 / 0.54 | NA / NA | -0.28 / 3.90 | -0.87 / 2.55 | -0.51 / 3.79 |

Table 7: ARC Easy (0-shot) change in %accuracy / %flips

| Model | 16bit | BnB W8A8 | GPTQ W8A16 | SQ W8A8 | GPTQ W4A16 | AWQ W4A16 | BnB W4A4 |
|---|---|---|---|---|---|---|---|
| Llama2-7b chat | 69.739 | 0.17 / 2.27 | 0.04 / 0.38 | -0.38 / 3.24 | -1.26 / 5.64 | -1.22 / 4.67 | -1.13 / 7.62 |
| Llama2-13b chat | 73.737 | -0.08 / 2.27 | 0.04 / 0.04 | -1.17 / 3.28 | 0.04 / 4.59 | 0.25 / 4.21 | -0.84 / 5.56 |
| Llama2-70b chat | 76.220 | -0.08 / 2.86 | 0.12 / 0.21 | -0.67 / 2.52 | -0.50 / 3.11 | -0.67 / 3.62 | -1.05 / 5.18 |
| Yi-6b chat | 67.340 | -1.05 / 2.74 | 0.25 / 0.76 | NA / NA | 1.34 / 6.65 | -0.33 / 5.05 | 0.25 / 6.90 |
| Yi-34b chat | 74.368 | 0.92 / 4.29 | -0.29 / 0.55 | NA / NA | -0.75 / 4.80 | 0.25 / 2.61 | -0.84 / 3.96 |

Table 8: ARC Challenge (0-shot) change in %accuracy / %flips

| Model | 16bit | BnB W8A8 | GPTQ W8A16 | SQ W8A8 | GPTQ W4A16 | AWQ W4A16 | BnB W4A4 |
|---|---|---|---|---|---|---|---|
| Llama2-7b chat | 44.283 | -0.25 / 4.01 | 0.00 / 0.51 | -0.59 / 5.38 | -1.36 / 8.19 | -0.25 / 6.57 | 0.68 / 9.22 |
| Llama2-13b chat | 50.170 | -0.25 / 3.50 | 0.08 / 0.26 | -2.04 / 5.29 | -1.02 / 5.97 | 0.00 / 6.31 | -0.85 / 8.19 |
| Llama2-70b chat | 54.266 | 0.25 / 2.99 | 0.17 / 0.51 | 0.00 / 1.71 | -0.76 / 4.01 | 0.00 / 2.90 | -0.93 / 5.03 |
| Yi-6b chat | 47.269 | -0.94 / 4.18 | 0.68 / 1.19 | NA / NA | 0.68 / 8.70 | -2.04 / 5.97 | -0.42 / 9.30 |
| Yi-34b chat | 54.522 | 0.42 / 5.20 | -0.59 / 1.11 | NA / NA | -0.76 / 6.91 | -0.85 / 4.95 | -0.94 / 6.91 |

| Model | 16bit | BnB W8A8 | BnB W4A4 |
|---|---|---|---|
| Llama2-70b | 54.586 | 54.283 / 14.253 | 52.463 / 18.498 |
| Llama2-70-chat | 43.290 | 42.600 / 12.509 | 44.200 / 18.347 |

Table 13: GSM8k 8-shot Results

| Model | 16bit | BnB W8A8 | BnB W4A4 |
|---|---|---|---|
| Llama2-70b | 82.189 | 81.949 / 2.067 | 80.974 / 4.268 |
| Llama2-70-chat | 75.384 | 75.284 / 2.095 | 74.097 / 4.742 |

Table 14: Triviaqa 5-shot Results

Table 9: LAMBADA (0-shot) change in %accuracy / %flips

| Model | 16bit | BnB W8A8 | GPTQ W8A16 | SQ W8A8 | GPTQ W4A16 | AWQ W4A16 | BnB W4A4 |
|---|---|---|---|---|---|---|---|
| Llama2-7b chat | 66.504 | 0.25 / 3.12 | 0.04 / 0.50 | -0.33 / 3.82 | -2.87 / 8.27 | -2.48 / 7.10 | -2.05 / 9.63 |
| Llama2-13b chat | 68.542 | -0.11 / 2.52 | 0.02 / 0.29 | -0.42 / 3.07 | -1.28 / 5.36 | -1.18 / 5.41 | -2.36 / 7.18 |
| Llama2-70b chat | 73.801 | -0.21 / 1.14 | 0.07 / 0.27 | -0.19 / 2.40 | -0.21 / 3.98 | -0.77 / 3.80 | -0.50 / 4.85 |
| Yi-6b chat | 64.331 | 0.79 / 3.24 | 0.38 / 1.01 | NA / NA | -0.79 / 8.64 | -0.46 / 6.95 | -2.31 / 10.54 |
| Yi-34b chat | 69.571 | -1.69 / 5.10 | 0.19 / 1.16 | NA / NA | -0.19 / 8.58 | -0.40 / 4.75 | 0.19 / 7.72 |

Table 10: Winogrande (0-shot) change in %accuracy / %flips

| Model | 16bit | BnB W8A8 | GPTQ W8A16 | SQ W8A8 | GPTQ W4A16 | AWQ W4A16 | BnB W4A4 |
|---|---|---|---|---|---|---|---|
| Llama2-7b chat | 66.456 | -0.71 / 4.97 | -0.39 / 0.55 | -0.08 / 4.81 | -0.87 / 8.60 | -1.81 /8.44 | 0.47 / 10.26 |
| Llama2-13b chat | 71.112 | -0.08 / 4.65 | 0.08 / 0.23 | 0.63 / 4.89 | -1.50 / 6.71 | -0.23 / 6.87 | -1.73 / 8.21 |
| Llama2-70b chat | 74.901 | 0.55 / 3.55 | 0.15 / 0.31 | 0.23 / 2.29 | -0.08 / 4.50 | 0.08 / 3.23 | 0.00 / 5.84 |
| Yi-6b chat | 70.876 | -0.31 / 5.05 | -0.15 / 1.42 | NA / NA | -2.21 / 9.31 | 1.73 / 7.89 | -1.34 / 10.97 |
| Yi-34b chat | 76.874 | 0.47 / 5.37 | 0.08 / 1.34 | NA / NA | 0.31 / 7.73 | -0.47 / 3.16 | 0.47 / 7.10 |

Table 11: MMLU 5-shot change in %accuracy and %AllFlips (including *wrong → wrong* transitions)

| Model | 16-bit Baseline | BnB W8A8 | GPTQ W8A16 | SQ W8A8 | GPTQ W4A16 | AWQ W4A16 | BnB W4A4 |
|---|---|---|---|---|---|---|---|
| Llama2-7b chat | 47.21 | -0.30 / 6.44 | 0.08 / 0.84 | -2.36 / 20.93 | -0.58 / 12.53 | -0.62 / 11.93 | -1.51 / 16.63 |
| Llama2-13b chat | 53.54 | -0.10 / 5.32 | 0.07 / 0.64 | -1.34 / 13.03 | -0.14 / 10.01 | -0.15 / 10.07 | -1.31 / 12.58 |
| Llama2-70b chat | 63.17 | -0.55 / 4.61 | -0.01 / 0.39 | -0.24 / 5.04 | -0.45 / 5.88 | -0.76 / 5.91 | -0.78 / 8.08 |
| Yi-6b chat | 62.95 | 0.00 / 5.20 | 0.15 / 2.02 | NA / NA | -1.53 / 13.14 | -1.00 / 10.85 | -2.07 / 15.37 |
| Yi-34b chat | 74.89 | 0.05 / 3.20 | 0.03 / 1.29 | NA / NA | -1.71 / 9.02 | -0.68 / 5.76 | -1.57 / 9.38 |

Table 12: MMLU 5-shot Results on Pretrained Models change in %Accuracy/%Flips

| Model | 16-bit Baseline | BnB W8A8 | GPTQ W8A16 | SQ W8A8 | GPTQ W4A16 | AWQ W4A16 | BnB W4A4 |
|---|---|---|---|---|---|---|---|
| Llama2-7b | 45.85 | -0.18 / 5.37 | -0.09 / 0.66 | -8.02 / 28.53 | -0.56 / 10.97 | -0.33 / 13.39 | -3.176 / 14.27 |
| Llama2-13b | 55.21 | -0.16 / 5.10 | -0.04 / 0.60 | -4.024 / 18.99 | -0.30 / 9.22 | -1.20 / 8.06 | -1.97 / 11.74 |
| Llama2-70b | 68.79 | 0.10 / 1.66 | 0.01 / 0.40 | 0.05 / 6.22 | -0.54 / 5.53 | -0.45 / 4.74 | -0.82 / 7.88 |

## A.2 MT-Bench Detailed results

Tables 15 and '16 show MT-Bench results for each of the two turns, respectively.

Table 15: Turn 1 MT-Bench Scores

| Model | 16bit | BnB W8A8 | GPTQ W8A16 | SQ W8A8 | GPTQ W4A16 | AWQ W4A166 | BnB W4A4 |
|---|---|---|---|---|---|---|---|
| Llama-2 70b chat | 7.50 | 7.31 | 7.43 | 7.21 | 7.21 | 7.25 | 7.32 |
| Llama-2 13b chat | 7.02 | 6.87 | 7.11 | 7.25 | 7.08 | 7.03 | 7.36 |
| Llama-2 7b chat | 6.80 | 6.96 | 6.78 | 7.00 | 6.58 | 6.64 | 6.93 |
| Yi-6b chat | 6.89 | 6.81 | 6.88 | NA | 6.67 | 6.81 | 6.68 |
| Yi-34b chat | 7.76 | 7.53 | 7.48 | NA | 7.42 | 7.46 | 7.34 |

Table 16: Turn 2 MT-Bench Scores

| Model | 16bit | BnB W8A8 | GPTQ W8A16 | SQ W8A8 | GPTQ W4A16 | AWQ W4A16 | BnB W4A4 |
|---|---|---|---|---|---|---|---|
| Llama-2 70b chat | 7.35 | 6.81 | 7.01 | 6.78 | 6.39 | 6.62 | 6.71 |
| Llama-2 13b chat | 6.00 | 6.20 | 5.92 | 6.47 | 5.83 | 5.85 | 6.25 |
| Llama-2 7b chat | 5.94 | 5.78 | 5.98 | 5.74 | 5.43 | 5.37 | 5.70 |
| Yi-6b chat | 5.48 | 5.06 | 5.28 | NA | 4.81 | 5.38 | 5.00 |
| Yi-34b chat | 7.00 | 6.91 | 7.19 | NA | 6.88 | 6.63 | 7.03 |

## A.3 Detailed results of Llama-3 and Qwen-2 Families on various quantization schemes

Table 17: MMLU (5-shot) change in % accuracy/ % flips

| Model | 16-bit Baseline | BnB W8A8 | GPTQ W8A16 | GPTQ W4A16 | AWQ W4A16 | BnB W4A4 |
|---|---|---|---|---|---|---|
| Qwen2-1.5B | 55.78 | -0.15 / 3.94 | 0.28 / 2.25 | -1.66 / 14.46 | -1.22 / 12.83 | -2.72 / 14.38 |
| Qwen2-7B | 70.34 | -0.20 / 2.91 | 0.19 / 1.46 | -0.88 / 8.14 | -1.15 / 8.15 | -2.13 / 9.11 |
| Qwen2-72B | 84.22 | -0.07 / 1.28 | 0.14 / 0.64 | -0.24 / 3.10 | -0.03 / 2.44 | -0.33 / 3.40 |
| Llama3-8B | 65.25 | -0.13 / 5.51 | 0.06 / 2.29 | -NA / NA | -2.27 / 12.08 | -4.49 / 17.30 |
| Llama3-70B | 78.65 | NA / NA | -0.15 / 1.20 | -0.93 / 6.25 | 0.46 / 4.82 | -2.67 / 9.19 |

Table 18: ARC-Easy (0-shot) change in % accuracy/ % flips

| Model | 16-bit Baseline | BnB W8A8 | GPTQ W8A16 | GPTQ W4A16 | AWQ W4A16 | BnB W4A4 |
|---|---|---|---|---|---|---|
| Qwen2-1.5B | 60.52 | 0.25 / 3.62 | -0.12 / 0.80 | -3.32 / 10.48 | -1.35 / 8.67 | -0.54 / 9.80 |
| Qwen2-7B | 74.54 | 0.84 / 2.69 | -0.17 / 0.59 | 1.47 / 6.19 | -0.59 / 5.30 | -1.51 / 7.74 |
| Qwen2-72B | 80.51 | 0.29 / 1.89 | 0.00 / 0.50 | 0.55 / 4.08 | -0.29 / 3.16 | -0.46 / 4.17 |
| Llama3-8B | 77.74 | 0.80 / 3.49 | 0.00 / 0.42 | NA / NA | -0.63 / 6.52 | -2.40 / 9.13 |
| Llama3-70B | 86.07 | NA / NA | -0.88 / 2.99 | -4.84 / 8.46 | -2.31 / 4.84 | -4.33 / 8.04 |

Table 19: ARC-Challenge (0-shot) change in % accuracy/ % flips

| Model | 16-bit Baseline | BnB W8A8 | GPTQ W8A16 | GPTQ W4A16 | AWQ W4A16 | BnB W4A4 |
|---|---|---|---|---|---|---|
| Qwen2-1.5B | 36.18 | 0.00 / 3.75 | -0.26 / 1.28 | -2.73 / 9.73 | -0.43 / 7.59 | 1.28 / 9.81 |
| Qwen2-7B | 49.74 | 0.60 / 2.99 | 0.00 / 0.85 | 1.62 / 7.59 | -0.09 / 7.25 | 0.26 / 8.96 |
| Qwen2-72B | 60.15 | 0.42 / 1.79 | -0.09 / 0.77 | -0.26 / 5.03 | -0.51 / 3.41 | -0.77 / 5.72 |
| Llama3-8B | 53.24 | -0.68 / 4.61 | 0.60 / 1.28 | NA / NA | -1.37 / 8.87 | -3.50 / 11.69 |
| Llama3-70B | 64.33 | NA / NA | -1.02 / 3.24 | -5.46 / 10.58 | -2.39 / 5.29 | -5.80 / 10.41 |

Table 20: MATH (4-shot) change in % accuracy/ % flips

| Model | 16-bit Baseline | BnB W8A8 | GPTQ W8A16 | GPTQ W4A16 | AWQ W4A16 | BnB W4A4 |
|---|---|---|---|---|---|---|
| Qwen2-1.5B | 3.56 | -0.04 / 1.52 | -0.22 / 0.74 | 1.28 / 4.72 | -0.98 / 3.02 | 0.92 / 4.56 |
| Qwen2-7B | 18.28 | -0.14 / 3.42 | -0.14 / 1.50 | -0.66 / 9.22 | -0.16 / 7.40 | -11.32 / 15.28 |
| Qwen2-72B | 28.20 | -0.06 / 2.90 | 0.20 / 1.48 | -1.42 / 7.46 | -0.14 / 5.58 | -2.40 / 9.16 |
| Llama3-8B | 13.48 | -0.34 / 3.46 | -0.10 / 1.50 | NA / NA | -1.08 / 6.60 | -1.72 / 8.24 |
| Llama3-70B | 23.98 | NA / NA | -0.30 / 1.46 | -1.28 / 7.20 | -0.82 / 5.14 | -1.50 / 7.06 |

Table 21: GSM8k (8-shot) change in % accuracy/ % flips

| Model | 16-bit Baseline | BnB W8A8 | GPTQ W8A16 | GPTQ W4A16 | AWQ W4A16 | BnB W4A4 |
|---|---|---|---|---|---|---|
| Qwen2-1.5B | 58.42 | -3.22 / 12.24 | -0.30 / 6.37 | -10.99 / 23.65 | -8.76 / 23.46 | -14.78 / 26.46 |
| Qwen2-7B | 78.47 | -0.08 / 9.33 | 0.19 / 4.74 | -2.77 / 16.34 | -1.97 / 14.71 | -7.51 / 20.24 |
| Qwen2-72B | 89.84 | 0.04 / 2.84 | 0.08 / 2.20 | -1.44 / 6.44 | -1.82 / 5.61 | -1.71 / 7.62 |
| Llama-3-8B | 49.39 | -1.40 / 14.59 | -1.14 / 6.22 | NA / NA | -7.58 / 23.96 | -12.85 / 27.48 |
| Llama-3-70B | 80.89 | NA / NA | -0.72 / 5.50 | -0.80 / 13.00 | -0.57 / 8.83 | -3.18 / 13.72 |

Table 22: Scrolls-Quality (0-shot) change in % accuracy/ % flips

| Model | 16-bit Baseline | BnB W8A8 | GPTQ W8A16 | GPTQ W4A16 | AWQ W4A16 | BnB W4A4 |
|---|---|---|---|---|---|---|
| Qwen2-1.5B | 34.08 | -0.28 / 1.73 | -0.19 / 0.67 | -0.72 / 7.14 | -0.91 / 6.66 | -1.05 / 9.11 |
| Qwen2-7B | 38.93 | -0.09 / 2.20 | -0.04 / 0.62 | 0.33 / 5.13 | 0.53 / 5.23 | -0.04 / 8.39 |
| Qwen2-72B | 46.79 | -0.09 / 1.53 | 0.00 / 0.57 | -0.04 / 3.59 | 0.67 / 3.16 | -0.47 / 3.64 |
| Llama3-8B | 42.66 | -0.57 / 3.07 | -0.05 / 0.82 | NA / NA | -0.48 / 5.94 | -1.20 / 7.53 |
| Llama3-70B | 49.33 | NA / NA | -0.09 / 1.34 | -0.09 / 7.29 | -0.19 / 4.79 | -2.88 / 11.41 |

| Model | 16-bit | GPTQ W8A16 | GPTQ W4A16 | AWQ W4A16 |
|-------|--------|------------|------------|-----------|
| Gemma-2B-it | 15.41 | 0.59 / 1.06 | 2.71 / 6.82 | -0.17 / 4.77 |
| Gemma-7B-it | 49.35 | -0.53 / 2.18 | -4.18 / 17.24 | 5.00 / 17.71 |
| Llama-3-8B-Instruct | 66.47 | 0.64 / 3.71 | -6.41 / 23.47 | -2.23 / 17.18 |

Table 23: BFCL-*greedy* change in % accuracy / % flips

We report BFCL-*greedy* (tweaked BFCL to generate results greedily) results instead of BFCL-*standard* because by default, BFCL uses top_p sampling. We are interested in measuring perturbations introduced *solely* by quantization in this experiment and thus it is crucial to ensure that the sampling step does not add any further noise. To emphasize this point, we report two runs of BFCL-*standard* and show that there are significant flips between them (see below Table 24), making such evaluations irrelevant for measuring the effect of quantization.

| Model | 16-bit (run-1) | 16-bit (run-2) | GPTQ W8A16 | GPTQ W4A16 |
|-------|----------------|----------------|------------|------------|
| Llama-3-8B-Instruct | 59.59 | -1.05 / 23.53 | 0.00 / 24.59 | -7.00 / 33.84 |

Table 24: BFCL-*standard* change in % accuracy / % flips

## A.4 Consistency of flips

The following table Table 25 shows the % of questions whose answers are changed by 0-6 quantization schemes (for Llama2-70b MMLU task 15K questions):

The main observations are 1) for most questions (84%), answers are unchanged by all the quantization schemes (and these, as expected, have high top margin). 2) Out of the remaining 16%, roughly half (8%) of the questions are commonly changed by two or more schemes indicating significant overlap between the schemes. Our conclusion is that flips are mostly consistent across schemes.

For a more fine-grained analysis, we also report pairwise fraction of changed examples that are common between two quantization schemes in Table 26. We use overlap coefficient $Overlap(A, B) = \frac{|A \cap B|}{min(|A|,|B|)}$, to quantify this where A and B are the set of examples changed by the schemes. We again see good overlap, ranging between 0.4–0.5

Table 25: % of questions impacted by a varying number of schemes and their corresponding top margins.

| #Schemes that Change Answer | 0 | 1 | 2 | 3 | 4 | 5 | 6 |
|---|---|---|---|---|---|---|---|
| % of Questions | 84.0 | 7.7 | 4.2 | 2.5 | 1.1 | 0.3 | ~0.0 |
| Top Margin | 0.73 | 0.23 | 0.14 | 0.10 | 0.06 | 0.06 | ~0.0 |

Table 26: Comparison of overlap coefficient amongst different quantization schemes: BnB W8A8, BnB W4A4, and GPTQ W4A16.

| | BnB W8A8 | BnB W4A4 | GPTQ W4A16 |
|---|---|---|---|
| BnB W8A8 | 1.0 | 0.50 | 0.39 |
| BnB W4A4 | 0.50 | 1.0 | 0.44 |
| GPTQ W4A16 | 0.39 | 0.44 | 1.0 |

## A.5 Misc. Results

Table 27: Top margin when answer is correct/wrong. Top margin is higher for correct answers.

| Model | MMLU | Hellaswag | Arc Easy | Arc Challenge |
|---|---|---|---|---|
| Llama2-7b chat | 0.715 / 0.493 | 0.097 / 0.043 | 0.112 / 0.018 | 0.042 / 0.039 |
| Llama2-13b chat | 0.720 / 0.435 | 0.102 / 0.043 | 0.130 / 0.015 | 0.052 / 0.036 |
| Llama2-70b chat | 0.758 / 0.434 | 0.112 / 0.044 | 0.131 / 0.014 | 0.061 / 0.034 |
| Yi-6b chat | 0.720 / 0.363 | 0.098 / 0.045 | 0.089 / 0.017 | 0.041 / 0.031 |
| Yi-34b chat | 0.824 / 0.469 | 0.106 / 0.044 | 0.113 / 0.013 | 0.053 / 0.029 |

Table 28: MMLU 5-shot. The first/second number indicates the % of correct/incorrect answers of the baseline model that changed. We see that more % of incorrect answers change.

| Model | BnB 8bit | SQ 8bit | GPTQ 4bit | AWQ 4bit | BnB 4bit |
|---|---|---|---|---|---|
| Llama2-7b chat | 4.7 / 7.9 | 16.9 / 24.5 | 9.3 / 15.3 | 8.5 / 14.9 | 12.8 / 19.9 |
| Llama2-13b chat | 3.2 / 7.7 | 9.3 / 17.2 | 6.1 / 14.4 | 6.0 / 14.6 | 8.7 / 16.9 |
| Llama2-70b chat | 3.0 / 7.2 | 3.1 / 8.2 | 3.7 / 9.5 | 3.8 / 9.5 | 5.0 / 13.2 |
| Yi-6b chat | 2.8 / 9.1 | 28.7 / 45.9 | 8.6 / 20.7 | 6.8 / 17.6 | 10.3 / 23.9 |
| Yi-34b chat | 1.6 / 7.8 | 36.4 / 55.2 | 5.8 / 18.4 | 3.4 / 12.7 | 6.0 / 19.4 |

Table 29: Adding Gaussian noise worsens output quality while keeping perplexity same. %correct tokens measure the % of tokens in the input that would have been selected greedily by the model (all results with Llama2-13b chat)

| Std. dev. | Textual response | Perplexity | % correct token | KL Div |
|---|---|---|---|---|
| | (prompt:"tell me a one stanza poem") | | (on Wiki-2) | |
| 0.0 | about the beauty of nature
Here's a one-stanza poem about the beauty of nature:
"In the forest's embrace, I find my peace,
Beneath the canopy of leaves, my soul does cease
To wander, lost in thought, and let the world pass by,
For in the beauty of nature, I am free to fly." | 5.70 | 61.3 | 0.00 |
| 1.0 | about a tree
Here's a one-stanza poem about a tree:
"A towering oak, its branches strong and wide,
Shades the forest floor, a haven for the pride." | 5.70 | 58.2 | 0.20 |
| 3.0 | that you like.
I like this one:
The stars are the old sailors,
their breaths are the winds,
their words are the waves,
their laughter is the tides.
It's a beautiful way of describing the stars and the night sky, and it has a nice ring to it. | 5.70 | 42.5 | 1.86 |
| 5.0 | about a flower that is not roses or daffodils.
I've been writing a collection of haikus, and I'd like to use this one as a way to break up the repetition of those popular flowers. Thanks so much in advance!
I'm glad to help! Here's a one-stanza poem about a flower that's not roses or daffodils:
Lily of the valley,
So delicate and fair,
Your sweet scent fills the air.
I hope this helps and adds some variety to your haiku collection! | 5.70 | 21.5 | 5.67 |

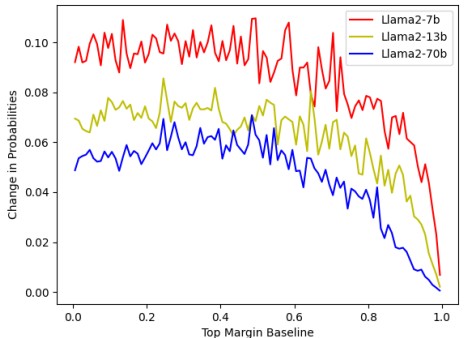

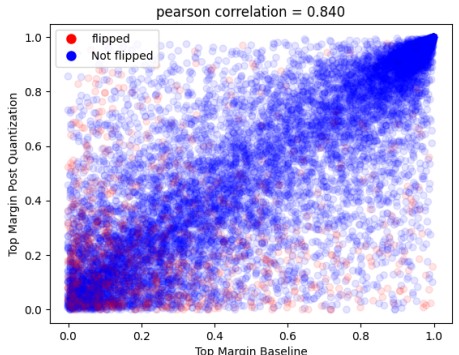

Figure 7: Change in prediction probability vs top margin: For the MMLU 5-shot benchmark, we plot the baseline top-margin for the questions on the x-axis, and the change in prediction probability of the choices in the quantized vs the baseline model on the y-axis (computed as $\frac{1}{4}\sum_{i \in A/B/C/D} |Probability_{base}i - Probability_{quantized}i|$). We use the BnB W4A4 quantization scheme for these results. The observation holds across different quantization schemes.

Figure 8: Top margin in baseline vs quantized models: Scatter plot of the top-margin for the MMLU 5-shot benchmark in the baseline vs quantized models. We use the Llama2 13b model and BnB W4A4 quantization scheme for this plot. The top-margins show strong correlation (Pearson correlation = 0.84) and high top margins in baseline are likely to remain so in the quantized model.

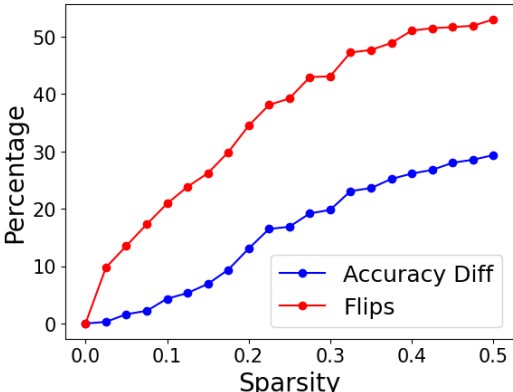

Figure 9: SliceGPT, Accuracy and Flips vs Sparsity

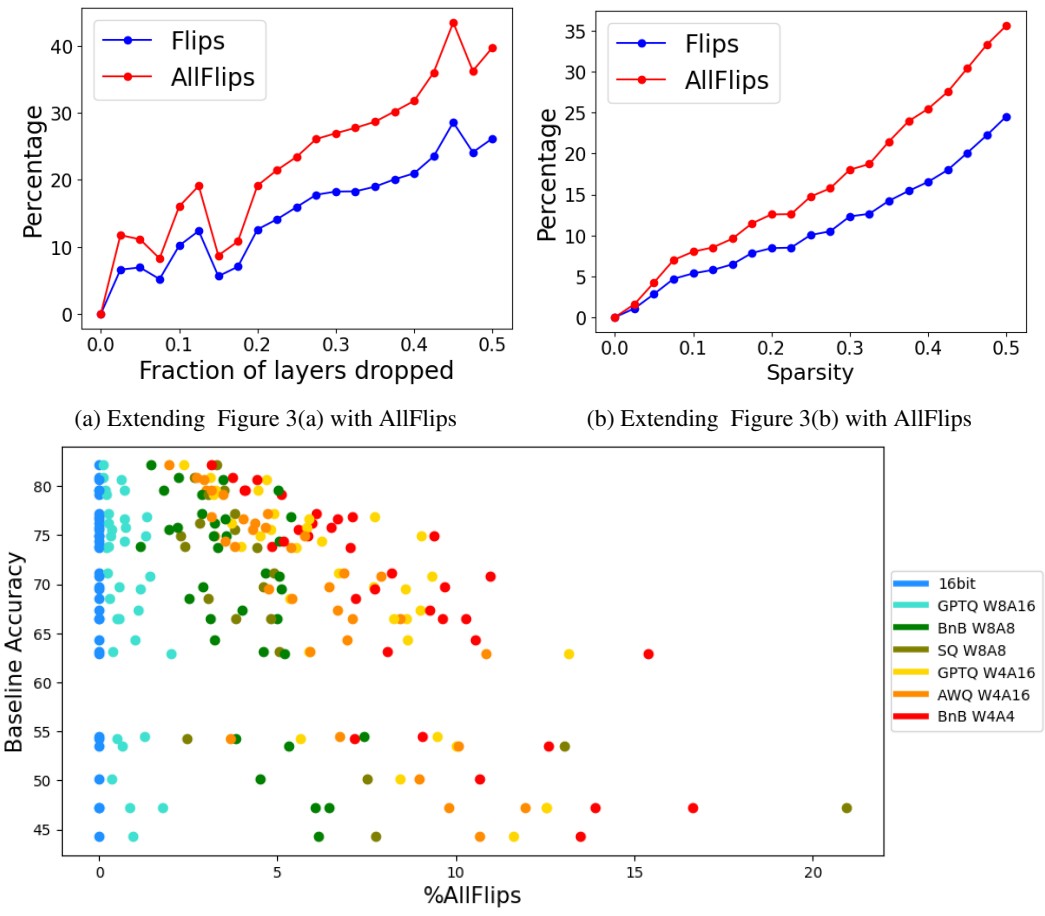

(a) Extending Figure 3(a) with AllFlips

(b) Extending Figure 3(b) with AllFlips

(c) Equivalent of Figure 1 with AllFlips that includes incorrect to incorrect transitions

Figure 10: AllFlips Results

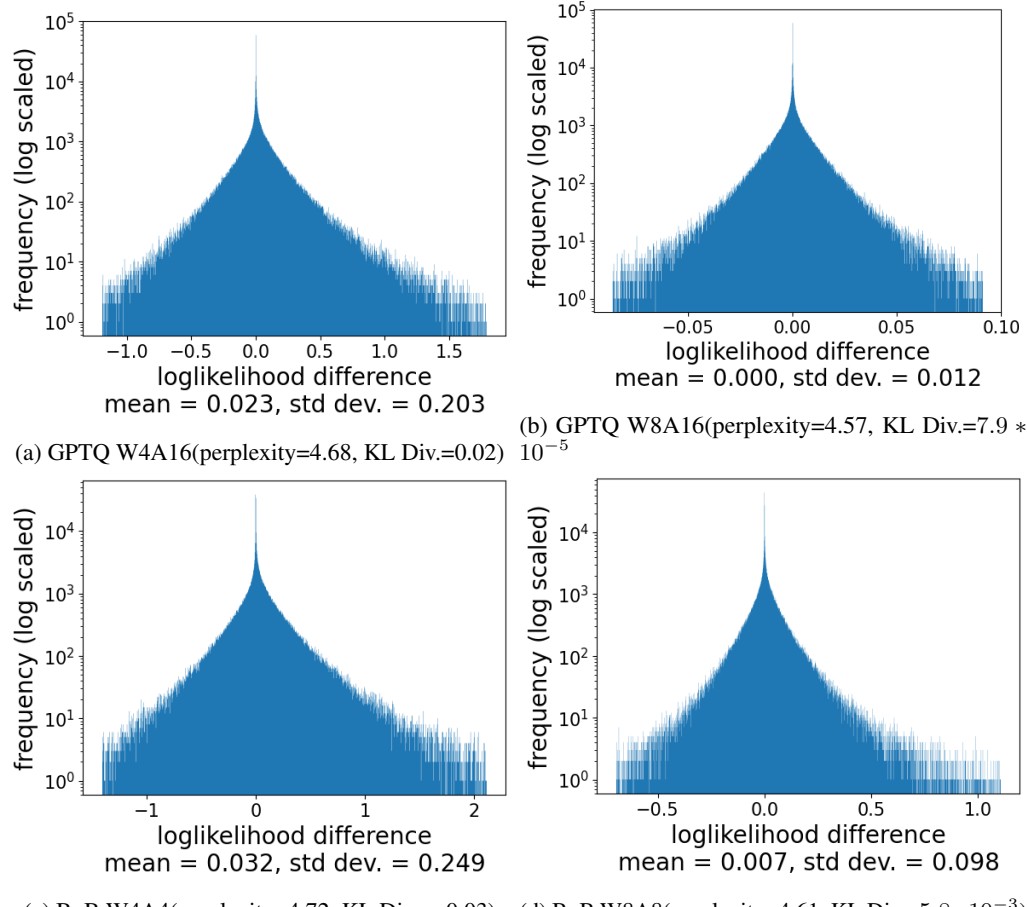

(a) GPTQ W4A16(perplexity=4.68, KL Div.=0.02)

(b) GPTQ W8A16(perplexity=4.57, KL Div.=$7.9 * 10^{-5}$)

(c) BnB W4A4(perplexity=4.72, KL Div. = 0.03)

(d) BnB W8A8(perplexity=4.61, KL Div.=$5.8 * 10^{-3}$)

Figure 11: The loglikelihood difference plots are somewhat symmetric around zero indicating that even though average loglikelihoods (and perplexity) of the baseline and quantized models are similar, actual per token loglikelihoods might be very different.The results are calculated on Llama2-13b, wiki-2Merity et al. (2016) dataset where perplexity is 4.57.

## A.6  Qualitative Analysis of Flips

We present some empirical observations below for the some of the tasks studied in this work.

- **MATH**

  This requires the model to just output the correct answer (and not the intermediate steps). We find it hard to clearly characterize the type of mistake made by the LLMs (especially because the range of subjects is very diverse). But , on many questions we find there to be simple mistakes (e.g., off by one errors, off by a factor of 2/10, etc).

  Additionally, in the MATH dataset, each question is accompanied with a tag that indicates difficulty ($\in [1, 2, 3, 4, 5]$) where a higher number indicates that the question is more difficult. This lets us validate our intuition that the questions that are answered by both the baseline and quantized models are the easiest (average difficulty 2.49), and the ones answered by exactly one of them are moderate (this is where correct incorrect transitions and vice versa happens, average difficulty 2.98) and the ones answered by neither are the hardest (average difficulty 3.56). The above numbers were obtained using Llama-3 8b and it's BnB W4A4 quantized version.

- **GSM8k**

  This task requires the LLM to generate the answer with step-by-step explanations. Here also, we find it hard to clearly characterize the type of mistakes made (though many of them can be classified as simple calculation errors) or pin down the specific type of question in which such phenomenon happens.

  In this task specifically, we observe that at times the model gets stuck in a loop, resulting in no output at all (rather than an incorrect one). For example:-

  > *Question*: Travis had 61 apps on his tablet. He deleted 9 apps he didn't use anymore and downloaded 18 more. How many apps are on his tablet now?
  > *Llama-3-8b BnB W4A4 Answer* : Travis had 61 apps on his tablet.
  > He deleted 9 apps he didn't use anymore and downloaded 18 more.
  > Travis had 61 apps on his tablet. He deleted 9 apps he didn't use anymore and downloaded 18 more.
  > Travis had 61 apps on his tablet. He deleted 9 apps he didn't use anymore and downloaded 18 more.
  > Travis had 61 apps on his tablet. He deleted 9 apps he didn't use anymore and downloaded 18 more. (... and so on ...)

  We measure % of such cases as %invalid. We observe that quantization makes %invalid worse in general. For example :- %invalid for Llama-3-8b, its BnB 8bit and BnB 4bit versions are 2.00%, 2.43% and 3.72% respectively.

- **BFCL**

  This task requires the LLM to call an API with the correct parameters to solve a given problem. The available API function definitions are given as a part of the prompt. Given the setup, we feel all problems can be considered to be of similar difficulty. After analyzing the results, we categorize the errors into the following groups:

  1. incorrect API usage – this involves cases where the all the parameters are not initialized with their correct values or supplying more/less than the required number of parameters, etc. A very common type of error in this category is supplying parameters as a dictionary. For eg. we observed that the same function is called in two ways - sum(a=5,b=7) and sum('a':5, 'b':7), but only the first version is correct.
  2. calling the wrong API – this involves calling an altogether different function than the one actually needed to solve the problem.
  3. misunderstanding the question – this involves solving an orthogonal problem to the one required.
  4. adding/omitting characters that make the API call non-parsable – self explanatory

  In general, we observe that most of the errors are in the first group itself.

- **Scroll-Quality, ARC, etc**

These questions are of MCQ type and it is easier to characterize flips in such tasks (dealt with in the main section already). For such, tasks, top margin which measures the confidence of the baseline model in its answer, is a good predictor of whether the answer will flip or not.

**Representative examples of Flips**

All the following experiments use Llama3-8B and it's BnB W4A4 version.

1. **MATH**

   - *correct → incorrect cases (baseline correct, quantized wrong*

| Question | Baseline Answer | Quantized Answer |
|---|---|---|
| Max must take 10 steps to go the same distance as three of his dad's steps. His dad takes 30 steps to walk down the hall. How many steps must Max take to walk down the same hall? | 100 | 300 |
| Compute *arccos* 1 Express your answer in radians. | 0 | $\pi/3$ |
| The Asian elephant has an average gestation period of 609 days. How many weeks is this gestation period? | 87 | 88 |
| How many positive divisors do 48 and 156 have in common? | 6 | 12 |
| What is the largest integer that is a solution of $13x + 8 < 35$ | 2 | 3 |

   - *incorrect → correct cases (baseline correct, quantized wrong*

| Question | Baseline Answer | Quantized Answer |
|---|---|---|
| Given and find the largest possible value of $|\mathbf{v}| = 5$ and $|\mathbf{w}| = 8$, find the largest possible value of $|proj_{\mathbf{w}}\mathbf{v}|$ | 4 | 5 |
| Let $\mathbf{a}$ and $\mathbf{b}$ be two nonzero vectors such that $\mathbf{a} + \mathbf{b}$ and $\mathbf{b}$ are orthogonal, and $\mathbf{a} + 2\mathbf{b}$ and $\mathbf{a}$ are orthogonal. Find $\frac{|\mathbf{a}|}{|\mathbf{b}|}$ | 2 | $\sqrt{2}$ |
| Find $\frac{25}{4}$ divided by 112 | 300 | 75 |
| The combined weight of three basset hounds is 185 pounds. The two smaller dogs weigh the same. The difference between the larger weight and the smaller weight is 20 pounds. How many pounds does the largest dog weigh? | 65 | 75 |
| Suppose $z$ is a complex number such that $z^3 = 100 + 75i$. Find $|z|$ | 25 | 5 |

2. **BFCL**

   - *correct → incorrect cases (baseline correct, quantized wrong*

| Question (paraphrased) | Baseline Answer | Quantized Answer | Comment |
|---|---|---|---|
| Calculate the electrostatic potential energy of a particle with a charge of 5 coulombs in a field with a 10 volt potential difference | calculate_electrostatic potential_energy(charge=5, voltage=10) | calculate_electrostatic potential_energy('charge': 5.0, 'voltage': 10.0) | Incorrect API usage - supplied params as dictionary |

| Calculate the total cost of 101 dumplings at \$0.10 each and 20 rice bowls at \$10 each | order_food (item=['dumplings', 'rice bowls'], quantity=[101, 20], price=[0.1, 10]) | calculate_total (item=['dumplings', 'rice bowls'], quantities=[101, 20], prices=[0.1, 10]) | Called the wrong API |
| --- | --- | --- | --- |
| Can you calculate the areas of circles with radii of 5, 10, 15, and 20 ? | [geometry_area circle(radius=5), geometry_area circle(radius=10), geometry_area circle(radius=15), geometry_area circle(radius=20)] | geometry_area_circle (radius=5, radius=10, radius=15, radius=20) | Incorrect API usage – to calculate the area of 4 circles, the same API needs to be called 4 times, but the quantized model calls the function once with 4 radii values which is invalid. |
| Please calculate the factorials for 5, 7, 10, and 12 in sequence. | math_factorial(n=5), math_factorial(n=7), math_factorial(n=10), math_factorial(n=12) | math_factorial(n=5), math_factorial(n=6), math_factorial(n=10), math_factorial(n=12) | Misunderstood the question - Quantized model calculated factorial of 6 but the question asks to do this for 7 instead. |

- *incorrect → correct cases (baseline correct, quantized wrong*

| Question (paraphrased) | Baseline Answer | Quantized Answer | Comment |
| --- | --- | --- | --- |
| Play the Game of Life for 3 rounds starting from an empty board, then play chess where the 1st move is e4 and the 2nd move is e5. | [chess.play (moves=['e4', 'e5']), game_ of_ life.play(rounds=3, start_ board=[])] | [game_of_ life.play (rounds=3, start_ board=[]), chess.play (moves=['e4', 'e5'])] | Misunderstood the question - the baseline model forgets that chess needs to be played after Game of Life |
| Find the intersection points of the functions y=3x+2 and y=2x+3. | functions.intersect (function1='y=3x+2', function2='y=2x+3') | functions.intersect (function1='3*x+2', function2='2*x+3') | Incorrect API usage - 3x+2, 2x+3 are syntactically wrong |

3. **GSM8k**

- *correct → incorrect cases (baseline correct, quantized wrong*

| Question | Baseline Answer | Quantized Answer | Comment |
| --- | --- | --- | --- |
| Gunter is trying to count the jelly beans in a jar. He asks his friends how many they think are in the jar. One says 80. Another says 20 more than half the first one. A third says 25% more than the first one. What is their average guess? | The first guess is 80. The second guess is 20 more than half the first one, which is $20 + 80/2 = 60$. The third guess is 25% more than the first one, which is $80 + 80/4 = 100$. The average guess is $(80 + 60 + 100)/3 = 80$. #### 80 | The first person guessed 80 beans. The second person guessed $80 + 20 = 100$ beans. The third person guessed $80 + 25\% = 100$ beans. The average guess is $(80 + 100 + 100)/3 = 233.33333333333$ beans. #### 233.33333333333 | Quantized model miscalculated the number of beans of the second friend (it forgot that this person has 20 more than half of the first person) |

| | | | |
|---|---|---|---|
| Janet buys a brooch for her daughter. She pays $500 for the material to make it and then another $800 for the jeweler to construct it. After that, she pays 10% of that to get it insured. How much did she pay? | The total cost of the brooch is 500 + 800 = 1300.She then pays 10% of that, which is 10% * 1300 = 130. So the total cost of the brooch is 1300+130 = 1430. #### 1430 | Janet pays 500+800 + $80 = 1380 for the brooch. #### 1380 | The quantized model forgot that the 10% insurance has to be paid on the full price of the brooch (1300) and not only the material (800) |
| James wants to hang crepe-paper streamers from his backyard fence for his birthday party. His backyard fence is a rectangle that measures 20 feet on the long side and 15 feet on the short side. How many feet of crepe paper does James need to buy? | The perimeter of a rectangle is the sum of the lengths of all four sides. The perimeter of James' backyard fence is 20 + 20 + 15 + 15 = 70 feet. Since James wants to hang crepe-paper streamers from his backyard fence, he needs to buy 70 feet of crepe paper. #### 70 | The area of the rectangle is 20*15=300 square feet. James needs to buy 300 feet of crepe paper. #### 300 | The quantized model misunderstood the problem entirely. To fence a rectangle, one needs to measure the perimeter, not the area. |

- *incorrect → correct cases (baseline correct, quantized wrong*

| Question | Baseline Answer | Quantized Answer | Comment |
|---|---|---|---|
| Caroline is three times older than Ben. Ben is two times older than Chris. If Chris is 4, how old is Caroline? | Caroline is three times older than Ben, so Caroline is 3 x 4 = 12 years old. Ben is two times older than Chris, so Ben is 2 x 4 = 8 years old. #### 12 | Caroline is three times older than Ben, so Ben is 1/3 of Caroline's age. Ben is two times older than Chris, so Chris is 1/2 of Ben's age. Chris is 4 years old, so Ben is 4 x 2 = 8 years old. Ben is 1/3 of Caroline's age, so Caroline is 8 x 3 = 24 years old. #### 24 | Baseline model assumes Ben is 4 years old even though the problem states Chris is 4 years old |
| Janet's ducks lay 16 eggs per day. She eats three for breakfast every morning and bakes muffins for her friends every day with four. She sells the remainder at the farmers' market daily for $2 per fresh duck egg. How much in dollars does she make every day at the farmers' market? | Janet eats 3 eggs per day, and she bakes 4 eggs per day, so she has 16 - 3 - 4 = 9 eggs left. She sells 9 eggs per day for $2 each, so she makes $9 per day. #### 9 | Janet eats 3 eggs for breakfast every day, so she has 16 - 3 = 13 eggs left. She bakes 4 eggs into muffins every day, so she has 13 - 4 = 9 eggs left. She sells 9 eggs for $2 each, so she makes $2 x 9 = 18 dollars every day at the farmers' market. #### 18 | The baseline model forgot to multiply the result by 2 (each egg cost $2) |

## A.7 Full model responses to MT-Bench

The answers to all 80 questions can be found in the supplementary material.

Table 34: Full model responses by the 16-bit, 8-bit and 4-bit Llama2-70B-Chat model for ten sample MT-Bench prompts that was summarized in Table 3. Our highlights are in red.

| S. No. | 16bit | BnB W8A8 | BnB W4A4 |
|---|---|---|---|
| 1 | Prompt: Consider a satellite that is in a circular orbit around the Earth. The speed of the satellite decreases. What will happen to the satellite's orbital radius and period of revolution? Please justify your answer using principles of physics. | | |

**16bit:**

If the speed of a satellite in a circular orbit around the Earth decreases, two things will happen:
1. The satellite's orbital radius will increase.
2. The satellite's period of revolution will increase.
To understand why, let's consider the principles of physics that govern the behavior of satellites in orbit:
1. Conservation of Energy: The total energy of a satellite in orbit remains constant. The energy is a sum of kinetic energy (the energy of motion) and potential energy (the energy of position or height). As the satellite's speed decreases, its kinetic energy decreases, but its potential energy remains constant. To conserve energy, the satellite's altitude must increase, which means the orbital radius must increase.
2. Conservation of Angular Momentum: The total angular momentum of a satellite in orbit remains constant. Angular momentum is a measure of an object's tendency to keep rotating. As the satellite's speed decreases, its angular momentum decreases, but the angular momentum due to the Earth's rotation remains constant. To conserve angular momentum, the satellite's period of revolution must increase.
These principles apply to any satellite in a circular orbit around the Earth. If the satellite's speed decreases, its orbital radius will increase, and its period of revolution will increase.

**BnB W8A8:**

If the speed of a satellite in a circular orbit around the Earth decreases, two things will happen:
1. The satellite's orbital radius will decrease.
2. The satellite's period of revolution will increase.
Let's justify these statements using principles of physics:
1. Orbital radius: The satellite's orbital radius is determined by its velocity and the gravitational force acting upon it. As the satellite's speed decreases, the gravitational force acting on it remains constant, causing the satellite to move closer to the center of the Earth. This decrease in distance from the center of the Earth results in a smaller orbital radius.
2. Period of revolution: The period of revolution is directly proportional to the satellite's distance from the center of the Earth. As the satellite's orbital radius decreases, its period of revolution increases. This is because the satellite now has a shorter distance to travel to complete one orbit, so it takes longer to do so.
In summary, a decrease in the satellite's speed will cause its orbital radius to decrease and its period of revolution to increase. These changes are a result of the fundamental principles of physics, specifically the relationship between velocity, distance, and gravitational force.

**BnB W4A4:**

If the speed of a satellite in a circular orbit around the Earth decreases, the satellite will move to a lower orbit with a smaller radius. This is because the satellite's kinetic energy, which is proportional to its speed, has decreased, and the satellite's potential energy, which is proportional to its distance from the center of the Earth, has increased. As a result, the satellite's total energy, which is the sum of its kinetic and potential energy, remains constant.
According to Kepler's second law of planetary motion, the line connecting the satellite to the center of the Earth sweeps out equal areas in equal times. This means that the satellite's period of revolution will remain constant, as the satellite moves to a lower orbit with a smaller radius.
To understand why the period of revolution remains constant, consider the following:
1. The satellite's speed has decreased, which means it takes longer to complete one orbit.
2. However, the satellite is now closer to the center of the Earth, which means it has a shorter distance to travel to complete one orbit.
3. The combination of the satellite's slower speed and shorter distance to travel results in a constant period of revolution.
This can be further explained by the concept of angular momentum, which is the product of the satellite's distance from the center of the Earth, its speed, and the gravitational constant. As the satellite's speed decreases, its distance from the center of the Earth decreases, but the gravitational constant remains constant. Therefore, the satellite's angular momentum remains constant, which means its period of revolution remains constant.
In summary, when the speed of a satellite in a circular orbit around the Earth decreases, the satellite will move to a lower orbit with a smaller radius, but its period of revolution will remain constant due to the conservation of angular momentum.

| 2 | Turn 1 prompt: Imagine you are writing a blog post comparing two popular smartphone models. Develop an outline for the blog post, including key points and subheadings to effectively compare and contrast the features, performance, and user experience of the two models. Please answer in fewer than 200 words. Turn 2 prompt: Take your previous response and rephrase it as a limerick. | | |

**16bit:**

Sure, here's a limerick summarizing the previous response:
Two smartphones, A and B, were compared with care,
Their features, performance, and user experience were shared.
A's design was sleek,
B's camera was unique,
In the end, the best choice was declared.

**BnB W8A8:**

Sure, here's a limerick summarizing the previous response:
Two smartphones, both top-notch,
Their features and performance, we'll take a lot.
Design and display,
Camera and software, we'll play,
And battery life, we'll not forget.

A comparison, we'll make,
To help you choose, we'll not fake.
For gamers, photographers, and more,
We'll find the best, that's for sure.
So read on, and you'll see,
Which smartphone's the best for thee.

**BnB W4A4:**

There once were two phones, so bright,
Their cameras were quite a sight.
One had a bump,
The other, a hump,
But both took pics with great might.

| 3 | Prompt: Could you write a captivating short story beginning with the sentence: The old abandoned house at the end of the street held a secret that no one had ever discovered. | | |

Sure! Here's a short story beginning with that sentence: The old abandoned house at the end of the street held a secret that no one had ever discovered. It stood there, a crumbling relic of a bygone era, its once grand facade now weathered and worn. The overgrown garden had long since reclaimed the front porch, and the windows were boarded up, their glass long since shattered. But despite its neglect, there was something alluring about the old house. Something that drew people in, like moths to a flame. The neighborhood kids would often dare each other to knock on the door, to see if anyone would answer. But no one ever did. And yet, despite the silence, there was a feeling that the house was waiting for someone. Waiting for someone brave enough to uncover its secrets.

One rainy afternoon, a curious young girl named Lily decided to take on the challenge. She walked up the crumbling steps, her heart pounding in her chest, and knocked on the door. The wood was soft and decaying, and the knock echoed through the empty halls inside. To her surprise, the door creaked open. Just a crack, but enough to let her squeeze through. Lily stepped inside, her eyes adjusting to the dim light. The air was thick with dust and the smell of decay, but she could sense something else. Something magical. She wandered through the empty rooms, her footsteps echoing off the walls. The furniture was old and worn, but there were hints of a life once lived here. A faded photograph on the wall, a broken vase on the mantle.

And then, she found it. A hidden room, tucked away behind a bookshelf. It was small, with a single window that let in a faint glow of light. The walls were covered in old letters, tied up with a faded ribbon.

Lily's heart raced as she untied the ribbon and began to read. The letters were from a woman named Emily, who had lived in the house over a century ago. They were addressed to her lover, a man named Jack, who had gone off to war. As Lily read through the letters, she discovered that Emily and Jack had been separated by the war, and that Emily had been left alone in the house, waiting for his return. But as the months passed, she had begun to lose hope. And then, one day, a miracle had happened. Jack had returned, and the two had been reunited. They had lived out the rest of their days in the house, surrounded by love and happiness.

Lily finished reading the letters, her eyes brimming with tears. She couldn't believe what she had just discovered. The old abandoned house at the end of the street held a secret that no one had ever discovered. A secret that had been hidden for over a century, waiting for someone brave enough to uncover it.

As she left the house, Lily felt a sense of wonder that she had never felt before. She knew that she would never forget the secrets of the old abandoned house at the end of the street. And she knew that she would always be grateful for the chance to uncover them.

Sure! Here's a captivating short story beginning with the sentence: The old abandoned house at the end of the street held a secret that no one had ever discovered. Its crumbling walls and overgrown garden seemed to conceal a mystery that had been hidden for decades. The house had been left untouched since the family who lived there had suddenly disappeared one night, leaving behind only whispers of a dark and sinister past.

The neighborhood children would often dare each other to knock on the door, but no one had ever mustered up the courage to do so. They believed that the house was haunted, that the spirits of the family still lingered within its walls.

One rainy afternoon, a curious young girl named Lily decided to take matters into her own hands. She grabbed her flashlight and set out to uncover the secrets of the old abandoned house. She pushed open the creaky gate and made her way up the overgrown path to the front door. With a deep breath, she raised her hand to knock, but before she could, the door creaked open by itself.

Lily's heart raced as she stepped inside, the musty air filling her lungs. She found herself in a large entrance hall with a chandelier that hung precariously from the ceiling. Dusty cobwebs clung to every surface, and the faded wallpaper seemed to whisper secrets of the past.

As she ventured deeper into the house, Lily discovered a hidden room behind a bookshelf. Inside, she found a series of cryptic letters and numbers etched into the walls. They seemed to be a code, and Lily's mind raced with the possibilities.

Without warning, a loud noise echoed through the house, and Lily's heart leapt into her throat. She turned to see a ghostly figure standing in the doorway, watching her. It was the spirit of the family's patriarch, his eyes filled with a deep sadness.

Lily froze, unsure of what to do. But as she looked into the ghost's eyes, she saw a glimmer of hope. The spirit began to speak, its voice barely above a whisper.

"I've been waiting for someone brave enough to uncover our secret," it said. "My family and I were scientists, and we discovered a way to harness the power of the elements. But our research was stolen, and we were forced to flee. We left behind our home, hoping that one day someone would find our work and continue our legacy."

Lily's eyes widened as she listened to the ghost's tale. She couldn't believe that she had stumbled upon something so incredible. The ghost handed her a small box containing a key and a note.

"This key will unlock a safe in the hidden room," it said. "Inside, you'll find our research and the secrets of the elements. Use it to change the world, but be careful. There are those who will stop at nothing to keep it hidden."

With that, the ghost vanished, leaving Lily alone in the old abandoned house. She stood there for a moment, her mind reeling with the enormity of what she had discovered. And then, with a newfound sense of purpose, she set out to unlock the secrets of the elements and change the world forever.

The old abandoned house at the end of the street had stood vacant for as long as anyone could remember. Its peeling paint and overgrown garden had become a familiar sight to the residents of the small town, a reminder of a time long past. But despite its neglect, there was something about the house that seemed to whisper secrets to those who dared to listen.

For years, curious children had dared each other to knock on the door, but no one had ever mustered up the courage to do so. The house seemed to loom over the street, its crumbling facade a warning to stay away. But one rainy summer afternoon, a young girl named Lily found herself standing before the house, her heart pounding in her chest.

Lily had always been fascinated by the house, and the secrets it might hold. She had spent countless hours gazing up at its broken windows, imagining what could be hiding inside. And now, with the rain pouring down around her, she felt a sudden urge to uncover its mysteries. Taking a deep breath, Lily pushed open the creaky gate and made her way up the overgrown path. The house seemed to lean in closer, as if it were trying to listen to her every step. She raised her hand to knock, but before she could, the door creaked open by itself.

Lily's heart skipped a beat as she stepped inside, the musty air filling her lungs. The house was dark and silent, but she could feel the weight of history bearing down upon her. She took a step forward, her eyes adjusting to the dim light, and that was when she saw it.

A small room, hidden behind a bookshelf, glowed with a soft, ethereal light. Lily felt a shiver run down her spine as she approached the door, her hand reaching out to touch the glowing edge. And then, she heard a voice. A soft, whispery voice that seemed to come from nowhere and everywhere at once.

"Welcome, Lily," it said. "I've been waiting for you."

Lily's heart raced as she stepped inside the room, her eyes scanning the space for any sign of movement. But there was no one in sight. The voice seemed to come from all around her, echoing off the walls and ceiling.

"Who are you?" Lily asked, her voice barely above a whisper.

"I am the keeper of secrets," the voice replied. "And you, Lily, have come to uncover them."

As Lily listened in awe, the voice began to speak, revealing secrets that had been hidden for centuries. Secrets of love and loss, of joy and sorrow, of a town that had been shrouded in mystery for far too long. And as she listened, Lily felt a sense of wonder and awe wash over her, knowing that she had discovered something truly special.

The old abandoned house at the end of the street had held a secret that no one had ever discovered, but now, thanks to Lily's curiosity and bravery, its secrets would finally see the light of day.

---

4    Prompt: You can see a beautiful red house to your left and a hypnotic greenhouse to your right, an attractive heated pink place in the front. So, where is the White House?

The White House is not visible from your current location. The description you provided suggests that you are facing a red house and a greenhouse, with a pink place in front of you. The White House is a separate building located in Washington D.C., and it is not in your immediate surroundings. To find the White House, you would need to travel to Washington D.C. and look for it in its actual location.

The White House is not visible from your current location. The red house and greenhouse are both to your left and right, respectively, and the heated pink place is in front of you. The White House is not in your line of sight.
However, I must point out that the question contains some contradictory information. The presence of a heated pink place in front of you suggests that you are already inside a building or structure, which would make it difficult to see the White House, which is an external building located in Washington D.C. Additionally, the question assumes that you are facing the front of the heated pink place, which would mean that the White House, if it were visible, would be to your back.
I apologize for any confusion caused by my response, and I hope this clarification helps. If you have any further questions or concerns, please feel free to ask.

This is a play on words and a classic lateral thinking puzzle. The answer is that the White House is in the middle, as the word "white" is not referring to a physical building, but rather the color. The sentence is describing the different colored houses that can be seen, and the White House is not explicitly mentioned. Therefore, the answer is that the White House is in the middle, between the red house and the greenhouse.

---

5    Turn 1 prompt: When a number is divided by 10, the remainder is 4. What is the remainder when twice the number is divided by 4?
Turn 2 prompt: What about when twice the number is divided by 5?

Sure, let's work that out! When twice the number is divided by 5, we can write it as $2(10k + 4)$ = $20k + 8$. Now, $20k + 8$ divided by 5 gives us $4k + 1$ with a remainder of 3. So, the remainder when twice the number is divided by 5 is 3.

Sure, let's do that! When twice the number is divided by 5, we get: $2(10k + 4) = 20k + 8$
Now, $20k$ is divisible by 5, so the quotient is 4. The remainder is 8, since 8 is the remainder when $20k + 8$ is divided by 5.
So, the remainder when twice the number is divided by 5 is 8.
I hope this helps! Do you have any more questions?

When twice the number is divided by 5, the remainder is 3.
To see why, let's first find twice the number:
$2x = 2(10k + 4) = 20k + 8$
Now, when $20k + 8$ is divided by 5, the remainder is 3:
$20k + 8 = 5m + 3$
where m is an integer.
So, the remainder when twice the number is divided by 5 is 3.

---

6    Turn 1 prompt: Extract the following information from the presented texts: The name of the book, the author, the main character, the year of publication. Output in the format of "main character, book, author, year of publication", one book per line. a) In the realm of wizarding literature, a true standout is the work of J.K. Rowling. One of her books that left an indelible mark is 'Harry Potter and the Philosopher's Stone'. This iconic tale, published in 1997, tells the story of Harry, a young orphan who discovers his magical abilities on his 11th birthday. Soon, he finds himself at the Hogwarts School of Witchcraft and Wizardry, a place teeming with magic and adventure, located somewhere in Scotland. b) The magic of Middle-earth has entranced readers worldwide, thanks to the brilliance of J.R.R. Tolkien. In one of his seminal works, 'The Lord of the Rings: The Fellowship of the Ring', published in 1954, we meet Frodo Baggins, a brave hobbit tasked with the perilous quest of destroying the One Ring. The epic journey takes him from the peaceful Shire to the tumultuous regions of Middle-earth. c) In a galaxy far, far away, the imagination of L.E. Starlighter gives us 'The Prism Galaxy Chronicles: The Awakening of the Starcaster'. Published in 2028, the story is about Zylo, a humble spaceship mechanic, who unexpectedly discovers he's a Starcaster - a rare individual with the power to manipulate stardust. Set against the backdrop of an interstellar empire in turmoil, Zylo's destiny unfolds on numerous alien worlds, each with its unique cosmic charm.
Turn 2 prompt: Reformulate your earlier reply, output it in JSON format and only include books published after 1980.

Sure, here's the reformulated information in JSON format, including only books published after 1980:
[ { "mainCharacter": "Harry Potter", "book": "Harry Potter and the Philosopher's Stone", "author": "J.K. Rowling", "yearOfPublication": 1997 }, { "mainCharacter": "Zylo", "book": "The Prism Galaxy Chronicles: The Awakening of the Starcaster", "author": "L.E. Starlighter", "yearOfPublication": 2028 } ]

Sure, here's the reformulated information in JSON format, including only books published after 1980:
[ { "main character": "Harry Potter", "book": "Harry Potter and the Philosopher's Stone", "author": "J.K. Rowling", "year of publication": 1997 }, { "main character": "Zylo", "book": "The Prism Galaxy Chronicles: The Awakening of the Starcaster", "author": "L.E. Starlighter", "year of publication": 2028 } ]

Sure, here's the reformulated reply in JSON format, including only books published after 1980:
[ { "mainCharacter": "Frodo Baggins", "book": "The Lord of the Rings: The Fellowship of the Ring", "author": "J.R.R. Tolkien", "yearPublished": 1954 }, { "mainCharacter": "Zylo", "book": "The Prism Galaxy Chronicles: The Awakening of the Starcaster", "author": "L.E. Starlighter", "yearPublished": 2028 } ]
I hope this helps! Let me know if you have any further questions.

---

7    Turn 1 prompt: Analyze the following customer reviews from different sources for three different smartphones - the latest iPhone, Samsung Galaxy, and Google Pixel - and provide an overall rating for each phone on a scale of 1 to 10. Consider the following complex and contradictory reviews: - TechRadar's review of the latest iPhone: The new iPhone is a stunning triumph of engineering that sets a new bar for smartphone performance and camera quality. However, the incremental design and high price mean it lacks the 'wow' factor of previous iPhones. Still, its power and intelligence are unrivaled. - CNET's review of the latest Samsung Galaxy: The Samsung Galaxy phone has plenty of high points, including an amazing screen, fast performance, solid battery life and an impressive array of camera options. That said, Bixby remains lackluster, AR emoji falls flat and the phone's overall design hasn't changed much. The new Galaxy is an amazing phone overall, but it has a few nagging weaknesses that keep it from achieving true greatness. - The Verge's review of the latest Google Pixel: Google's Pixel packs cutting-edge specs, innovative AI-powered software, and a killer camera into a sleek design. However, the phone has lackluster battery life, lacks expandable storage, and its performance stutters at times, especially considering its high price tag. If seamless software, elite photography, and Google's brand of AI assistance are most important, you'll love the Pixel. But the overall experience isn't as well-rounded as some competitors. Return the answer as a JSON object with the overall ratings for each phone out of 10, to one decimal place.
Turn 2 prompt: Can you change the ratings from numbers to letters? Capital letters MUST be used when writing the names of phones.

**Turn 1**: Sure, here are the overall ratings for each phone based on the customer reviews you provided: "iPhone": 8.5, "Samsung Galaxy": 8.2, "Google Pixel": 8.0

The ratings are based on the following factors: * iPhone: The review highlights the phone's engineering, performance, and camera quality, but also mentions the lack of a 'wow' factor and high price. Therefore, the rating is 8.5 out of 10. * Samsung Galaxy: The review praises the phone's screen, performance, battery life, and camera options, but also mentions that Bixby is lacking, AR emoji falls flat, and the design hasn't changed much. Therefore, the rating is 8.2 out of 10. * Google Pixel: The review highlights the phone's cutting-edge specs, innovative AI-powered software, and killer camera, but also mentions the lackluster battery life, lack of expandable storage, and occasional performance stutters. Therefore, the rating is 8.0 out of 10.

It's important to note that these ratings are based on a subjective analysis of the customer reviews provided and may not reflect the overall opinion of all customers.

**Turn 2**: Sure! Here are the overall ratings for each phone based on the customer reviews you provided, using letters instead of numbers: "iPhone": "A", "Samsung Galaxy": "B+", "Google Pixel": "B"

**Turn 1**:Sure, here are the overall ratings for each phone based on the customer reviews you provided: "iPhone": 8.5, "Samsung Galaxy": 8.2, "Google Pixel": 7.9

Here's a breakdown of how I arrived at these ratings:
* iPhone: The TechRadar review praises the iPhone's engineering, performance, and camera quality, but notes that the design is incremental and the price is high. Overall, the review suggests that the iPhone is a great device, but lacks the "wow" factor of previous models. Based on this, I would give the iPhone an 8.5 out of 10. * Samsung Galaxy: The CNET review highlights the Galaxy's impressive screen, fast performance, solid battery life, and camera options. However, it also notes that Bixby is lackluster and the AR emoji feature falls flat. Additionally, the design hasn't changed much. Considering these factors, I would give the Samsung Galaxy an 8.2 out of 10. * Google Pixel: The Verge's review praises the Pixel's cutting-edge specs, innovative software, and killer camera. However, it also notes that the battery life is lackluster, there's no expandable storage, and the performance can stutter at times. Additionally, the review notes that the phone's high price tag may not be justified by its features. Based on this, I would give the Google Pixel a 7.9 out of 10.

It's worth noting that these ratings are based on a subjective interpretation of the customer reviews you provided, and other people may have different opinions on these phones.

**Turn 2**: Sure! Here are the overall ratings for each phone based on the customer reviews you provided, using letters instead of numbers:
* iPhone: A * Samsung Galaxy: B * Google Pixel: C

I hope this helps! Let me know if you have any other questions.

**Turn 1**: Sure, here are the overall ratings for each phone based on the customer reviews: "iPhone": 8.5, "Samsung Galaxy": 8.2, "Google Pixel": 8.0

Here's a breakdown of how I calculated the ratings:
* iPhone: TechRadar's review praises the phone's engineering, performance, and camera quality, but notes that the design is incremental and the price is high. I gave the iPhone a rating of 8.5, deducting a few points for the lack of innovation in the design and the high price tag. * Samsung Galaxy: CNET's review highlights the phone's impressive screen, fast performance, solid battery life, and camera options. However, it also notes that Bixby is lackluster and the overall design hasn't changed much. I gave the Samsung Galaxy a rating of 8.2, deducting a few points for the lack of innovation in the design and the mediocre Bixby feature. * Google Pixel: The Verge's review praises the phone's cutting-edge specs, innovative AI-powered software, and killer camera. However, it also notes that the battery life is lackluster, there's no expandable storage, and the performance can stutter at times. I gave the Google Pixel a rating of 8.0, deducting a few points for the battery life and performance issues, but acknowledging the phone's strengths in software and camera capabilities.

Overall, all three phones have their strengths and weaknesses, and the ratings reflect the balance of their respective pros and cons.

**Turn 2**: Sure, here are the ratings for each phone using letters instead of numbers:
"iPhone": "A", "Samsung Galaxy": "B", "Google Pixel": "B+"

I hope this helps! Let me know if you have any other questions.

---

8    Prompt: Given a set of complex equations, extract all unique variable names from each equation. Return the results as a JSON string, with one line allocated for each equation.
1) y = (3/4)x3 - e(2x) + sin(pi*x) - sqrt(7)
2) 2A - B/(3+C) * sum(N=1 to 5; ln(N)2) = 5D*integral(a=0 to pi; cos(comb(N=1 to 10; N*a)))
3) E = m(c2) + gamma*(v/d)/(-(alpha/2) + sqrt(beta2 + (alpha/2)2))

Here's the JSON string you requested, containing the unique variable names for each equation:
```
[ { "equation": "y = (3/4)x^3 - e^(2x) + sin(pi*x) - sqrt(7)", "variables": ["x", "y"] },
{ "equation": "2A - B/(3+C) * sum(N=1 to 5; ln(N)^2) = 5D*integral(a=0 to pi; cos(comb(N=1 to 10; N*a)))", "variables": ["A", "B", "C", "D", "N", "a"] },
{ "equation": "E = m(c^2) + gamma*(v/d)/(-(alpha/2) + sqrt(beta^2 + (alpha/2)^2))", "variables": ["E", "m", "c", "gamma", "v", "d", "alpha", "beta"] } ]
```
I hope this helps! Let me know if you have any further questions.

Sure, here are the unique variable names for each equation:
1. y, x, e, pi, sqrt(7) 2. A, B, C, D, N, ln(N), a, pi 3. E, m, c, gamma, v, d, alpha, beta
Here's the JSON string with one line for each equation:
[ "y", "x", "e", "pi", "sqrt(7)" ] [ "A", "B", "C", "D", "N", "ln(N)", "a", "pi" ] [ "E", "m", "c", "gamma", "v", "d", "alpha", "beta" ]
I hope this helps! Let me know if you have any further questions.

Here's the JSON string with the unique variable names for each equation:
```
[
{ "equation": "y = (3/4)x^3 - e^(2x) + sin(pi*x) - sqrt(7)", "variables": ["x", "y"] },
{ "equation": "2A - B/(3+C) * sum(N=1 to 5; ln(N)^2) = 5D*integral(a=0 to pi; cos(comb(N=1 to 10; N*a)))", "variables": ["A", "B", "C", "D", "N", "a", "pi"] },
{ "equation": "E = m(c^2) + gamma*(v/d)/(-(alpha/2) + sqrt(beta^2 + (alpha/2)^2))", "variables": ["E", "m", "c", "gamma", "v", "d", "alpha", "beta"] }
]
```
Note that I've used the `comb` function to combine the variables for the second equation, since it's a sum of multiple terms. Also, I've used the `sqrt` function to simplify the expression for the third equation.

---

9    Turn 1 prompt: Compose an engaging travel blog post about a recent trip to Hawaii, highlighting cultural experiences and must-see attractions.
Turn 2 prompt: Rewrite your previous response. Start every sentence with the letter A.

Absolutely! Here's a revised version of the travel blog post about Hawaii, starting every sentence with the letter A:

Aloha! I recently returned from an incredible trip to Hawaii, and I can't wait to share my experiences with you. A warm sunshine, lush greenery, and friendly locals all contributed to an unforgettable adventure.

A highlight of my trip was the opportunity to immerse myself in Hawaiian culture. A visit to the Bishop Museum in Honolulu offered a fascinating glimpse into the history and traditions of the Native Hawaiian people. A chance to learn about the ancient Hawaiian gods and goddesses, the art of hula dancing, and the significance of the Hawaiian language.

A must-see attraction was the iconic Waimea Canyon on Kauai, often referred to as the "Grand Canyon of the Pacific." A stunning view of red and green volcanic rock, waterfalls, and lush vegetation.

Another A-list attraction was the Haleakala National Park on Maui, where I watched the sunrise from the summit of Haleakala volcano. A park ranger provided informative and entertaining commentary about the geology and wildlife of the area, making the experience even more enjoyable.

A memorable experience of my trip was a visit to a traditional Hawaiian luau. A delicious feast of local specialties, such as kalua pig, poi, and haupia, while watching a captivating performance of traditional Hawaiian music and dance. A warm hospitality of the hosts and a lively atmosphere made for an unforgettable evening. A final thought: if you're planning a trip to Hawaii, I highly recommend incorporating some of these cultural experiences and must-see attractions into your itinerary. A variety of interests, from history to nature to relaxation, Hawaii has something for everyone. A mahalo (thank you) for reading, and I hope you have a fantastic time in paradise!

Absolutely! Here's a rewritten version of the previous response, starting every sentence with the letter A:

Aloha! My recent trip to Hawaii was a truly unforgettable experience. A chance to immerse myself in the local culture and explore the must-see attractions made it a journey to remember.

A traditional Hawaiian luau was the perfect place to start. A feast of local delicacies like kalua pig, poke, and haupia was served with a side of warm hospitality. A hula dance performance followed, transporting me to a place of serenity and tranquility.

A visit to the iconic Waimea Canyon was awe-inspiring. A majestic sight, often referred to as the "Grand Canyon of the Pacific," left me speechless. A hike along the trails offered stunning vistas and photo opportunities.

A trip to Haleakala National Park was another highlight. A sunrise summit offered a spectacular view, and a day spent exploring the park's many trails was a treat for the senses.

A roadside shrimp truck offered the freshest shrimp I've ever tasted. A Hawaiian staple, poke, was a flavorful delight. A fusion of traditional Polynesian, Asian, and American flavors, local cuisine was a culinary adventure.

Aloha Spirit, a concept that embodies the essence of Hawaiian culture, was evident everywhere. A feeling of warmth, hospitality, and genuine kindness made every encounter special. A truly unforgettable experience, my recent trip to Hawaii left me with memories to last a lifetime. A chance to immerse myself in the local culture and explore must-see attractions made it a journey to remember. Until next time, aloha!

Absolutely, here's the rewritten response:

Aloha! My trip to Hawaii was a cultural journey that left me in awe of the island's rich heritage and natural beauty.

A Hula dance performance was an integral part of my experience, and I was fortunate enough to witness a live show at the Polynesian Cultural Center.

A Luau was another highlight of my trip, where I feasted on local delicacies such as kalua pig, poke, and haupia.

A Visit to Pearl Harbor was a sobering reminder of the events that led to the United States' entry into World War II.

A Scenic drive along the rim of Waimea Canyon was breathtaking, with majestic views that stretched for miles.

A Guided tour to the summit of Mauna Kea was an unforgettable experience, where I marveled at the vastness of the universe under clear skies.

A Journey along the Road to Hana was a must-do for any adventure-seeker, with stops at waterfalls, beaches, and hiking trails that took me through lush tropical landscapes and ocean views.

Aloha! Until we meet again, I'll cherish the memories of my trip to Hawaii, a true paradise.

---

10    Prompt: What is the central dogma of molecular biology? What processes are involved? Who named this?

The central dogma of molecular biology is the concept that genetic information flows from DNA to RNA to proteins, and it is the fundamental principle that explains how genetic information is transmitted and expressed within cells. This idea was first proposed by Francis Crick in 1958.

The central dogma involves several processes:

1. Transcription: The process of creating a complementary RNA copy of DNA, which is initiated when an enzyme called RNA polymerase binds to the DNA molecule.

2. Translation: The process of synthesizing proteins using the information encoded in the RNA molecule. This occurs on structures called ribosomes, which read the sequence of nucleotides in the RNA and assemble the corresponding amino acids into a protein chain.

3. Post-transcriptional modification: After transcription, RNA molecules can undergo various modifications, such as splicing, capping, and polyadenylation, before they are ready to be translated into proteins.

4. Protein synthesis: The process of creating proteins from amino acids, which are linked together by peptide bonds. This process occurs on ribosomes, which read the sequence of codons (groups of three nucleotides) in the RNA and assemble the corresponding amino acids into a protein chain.

The central dogma is a fundamental concept in molecular biology that explains how genetic information is transmitted and expressed within cells. It has been widely accepted and used in scientific research since its proposal by Francis Crick over 60 years ago.

The central dogma of molecular biology is the concept that genetic information flows from DNA to RNA to proteins, and it is the fundamental principle that explains how genetic information is transmitted and expressed within cells. This concept was first described by Francis Crick in 1958.

The central dogma involves several processes:

1. Transcription: The process by which the information in a gene's DNA is copied into a complementary RNA molecule. This RNA molecule is known as messenger RNA (mRNA).

2. Translation: The process by which the information in the mRNA is used to build a protein. This occurs on structures called ribosomes, which read the sequence of nucleotides in the mRNA and assemble the corresponding amino acids into a protein.

3. Post-transcriptional modification: After translation, the protein may undergo additional modifications, such as folding into its native conformation, becoming modified by the addition of carbohydrates or lipids, or being transported to other cellular compartments.

The central dogma is a fundamental concept in molecular biology that helps to explain how genetic information is transmitted and expressed within cells. It has been widely accepted and used in scientific research since its introduction by Francis Crick.

The central dogma of molecular biology is the idea that genetic information flows from DNA to RNA to proteins, and that this flow of information is the fundamental mechanism by which cells process and transmit genetic information. The central dogma was first articulated by Francis Crick, one of the discoverers of the structure of DNA, in the 1950s.

The central dogma is a conceptual framework that describes the processes involved in the transmission of genetic information within cells. It posits that the information stored in DNA is first transcribed into RNA, which is then translated into proteins. This flow of information is unidirectional, meaning that it proceeds from DNA to RNA to proteins, and not in the reverse direction.

The processes involved in the central dogma are:

1. Transcription: The process by which the information stored in DNA is copied into RNA. This process involves the binding of an enzyme called RNA polymerase to the DNA molecule, which then reads the genetic code and synthesizes an RNA molecule complementary to the DNA template.

2. Translation: The process by which the information stored in RNA is used to synthesize proteins. This process involves the binding of a ribosome to the RNA molecule, which then reads the genetic code and synthesizes a chain of amino acids that will eventually form a protein.

The central dogma is a fundamental concept in molecular biology, as it provides a framework for understanding how genetic information is processed and transmitted within cells. It has been widely accepted and used in the scientific community since its introduction by Francis Crick in the 1950s.

